# Causal Influence Aware Counterfactual Data Augmentation

## Abstract

Pre-recorded data and human-collected demonstrations are both valuable and practical resources for teaching robots complex behaviors. Ideally, learning agents should not be constrained by the scarcity of available demonstrations, but rather generalize to as many new situations as possible. However, the combinatorial nature of real-world scenarios typically requires a huge amount of data to prevent neural network policies from picking up on spurious and non-causal factors. We propose CAIAC, a data augmentation method that can create feasible synthetic samples from a fixed dataset without the need to perform new environment interactions. Motivated by the fact that an agent may only modify the environment through its actions, we swap causally *action*-unaffected parts of the state-space from different observed trajectories in the dataset. In high-dimensional benchmark environments, we observe an increase in generalization capabilities and sample efficiency.

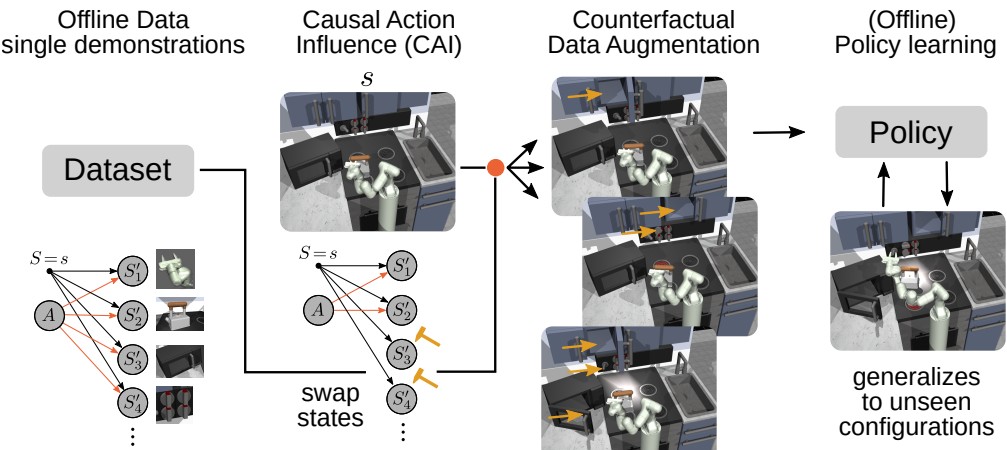

Figure 1: Overview of the proposed approach. Interactions between the agent and entities in the world are sparse. We use causal action influence (CAI), a local causal measure, to determine action-independent entities and create counterfactual data by swapping states of these entities from other observations in the dataset. Offline learning with these augmentations leads to better generalization.

## 1 Introduction

Teaching robots via demonstrations and collected datasets is a promising route towards capable robotic assistants (Bahl et al., 2022; Brohan et al., 2022; 2023). However, this approach encounters a fundamental challenge when applied to realistic scenarios, due to the combinatorial complexity spurring from the existence of many entities (Battaglia et al., 2018). The core of the problem is that demonstrations can only cover a small fraction of the vast array of possible configurations, leaving robots unable to robustly generalize to unfamiliar situations.

Let us consider how one may teach a robot to perform various tasks in a kitchen. When demonstrating several kitchen-related activities, the operation for each tool and appliance may be shown separately: opening doors to retrieve plates, sliding drawers to obtain tools, cutting vegetables, operating the

microwave — a list of elementary activities can potentially be very long. With sufficient data for each of them, increasingly powerful offline learning methods can be applied to robots, and enable them to perform these tasks under the demonstrated conditions.

However, robots still tend to fail when exposed to slight changes in their environment. In this case, the operation of the robot could be jeopardized by simply changing seemingly unrelated aspects of the kitchen, such as leaving one of the drawers open. In general, learning algorithms are notoriously prone to picking up on spurious correlations in the data, such as the fact that all drawers in the motivating kitchen example would be typically closed during demonstrations. In contrast, humans are remarkably good at inferring what parts of the environment are relevant to solve a task, possibly due to relying on a causal representation of the world (Pearl & Mackenzie, 2018).

To address this shortcoming in current methods, we propose an approach rooted in a local causal inference perspective. By examining the causal relationships between actions and objects in a specific context, we aim to empower robots with the ability to reason and adapt in complex environments. Our approach introduces counterfactual data augmentations without the need for additional environment interactions, nor relying on counterfactual model rollouts. Instead, we exploit recorded data by substituting locally causal independent factors with those of different observed trajectories.

The idea of creating counterfactual data augmentations has recently been pursued in Pitis et al. (2020), in which a heuristic, leveraging the attention weights of a transformer world model, is used to determine influences among objects and between agent and objects. Estimating the entire causal structure remains however a challenging task, and is in general prone to misinformations, particularly if attempted from offline data. Taking this into account, we exploit the assumption that an agent can only affect its environment through actions, and deem action-influence to be more important for policy learning than potential object-object interactions. By partially trading off generality, this inductive bias on the underlying causal structure reduces the problem of estimating the full causal structure to only measuring the influence of actions over objects. This quantity can then be explicitly estimated through the local Causal Action Influence (CAI) measure proposed by Seitzer et al. (2021), thus removing the need for heuristics for causal discovery. In practice, the measure can be computed by training a transition model to approximate state-conditioned mutual information (Cover, 1999). As our approach only evaluates causal influence on observed data, the transition model only needs to make locally accurate predictions about the connection between actions and objects. Nevertheless, the resulting counterfactual augmentation we propose, provides global coverage and generates training data that would otherwise be far out-of-distribution, as illustrated in Fig. 1.

Our framework works as an independent module and can be used with any learning algorithm. We demonstrate this through empirical results in high-dimensional offline goal-conditioned tasks, applying our method to fundamentally different data distributions and learning methods. Namely, we couple our method with offline goal-conditioned skill learning on the Franka-Kitchen environment (Gupta et al., 2019), and classical offline goal-conditioned reinforcement learning on Fetch-Push with two cubes (Andrychowicz et al., 2017). In both cases, we show that our method, which we refer to as *Causal Influence Aware Counterfactual Data Augmentation* (CAIAC) leads to enhanced generalization and improved performance when learning from a modest amount of demonstrations.

## 2 BACKGROUND

Markov Decision Processes (MDPs) are typically used as the basic semantics for optimal planning in stochastic environments. They are described by the tuple $(\mathcal{S}, \mathcal{A}, P, R, \gamma)$, consisting of state space, action space, transition kernel, reward function and discount factor, respectively. In this framework, the environment is modeled via a set of states $s \in \mathcal{S}$, which evolve stochastically according to the transition kernel $P$.

High-dimensional state spaces can be generally decomposed into a series of entities that interact with each other. In this paper, we model this by assuming a known and fixed state-space factorization $\mathcal{S} = \mathcal{S}_1 \times ... \times \mathcal{S}_N$ for $N$ entities, where each factor $\mathcal{S}_i$ corresponds to the state of an entity. In practice, there are methods that allow to automatically determine the number of factors (Zaheer et al., 2017) and to learn latent representations of each entity (Burgess et al., 2019; Zadaianchuk et al., 2023)(Locatello et al., 2020; Greff et al., 2019; Jiang et al., 2019; Seitzer et al., 2022). While we do not consider them for simplicity, our method can be applied on top of such techniques.

## 2.1 CAUSAL GRAPHICAL MODELS

We may model the state evolution of the underlying MDP from time $t$ to $t + 1$ using a causal graphical model (CGM) (Peters et al., 2017; Pearl, 2009) over the set of random variables $\mathcal{V} = \{S_1, ..., S_N, A, S'_1, ..., S'_N\}$. The causal graphical model consists of a directed acyclic graph (DAG) $\mathcal{G}$ and a conditional distribution $P(V_j \mid \mathrm{Pa}_{\mathcal{G}}(V_j))$ for each node $V_j \in \mathcal{V}$, where $\mathrm{Pa}_{\mathcal{G}}(V_j)$ denotes the set of parents of $V_j$ in the causal graph $\mathcal{G}$. For readability, we omit the time index $t$ and use $'$ to denote variables at time $t + 1$. We assume that the joint distribution $P_{\mathcal{V}}$ is Markov (Peters et al., 2017, Def. 6.21) with respect to the DAG $\mathcal{G}$, and factorizes as

$$p(v_1, ..., v_{|\mathcal{V}|}) = \prod_{j=1}^{|\mathcal{V}|} p(v_j \mid \mathrm{Pa}_{\mathcal{G}}(V_j)). \qquad (1)$$

Due to the structure of the MDP and our knowledge about time, we assume, as in (Seitzer et al., 2021; Pitis et al., 2020), a causal graph without connections among nodes at the same time step and there are no edges from the future to the past, as depicted in Fig. 2(a). If the graph $\mathcal{G}$ is structurally minimal, we can think of its edges as representing global causal dependencies. Each node is independent of its non-descendants given its parents, so that $S'_j \perp\!\!\!\perp V_j \mid \mathrm{Pa}(S'_j)$ for all nodes $V_j \notin \mathrm{Pa}(S'_j)$ (Peters et al., 2017, Def 6.21). The probability distribution of $S'_j$ is hence fully specified by its parents $P(S'_j|S, A) = P(S'_j \mid \mathrm{Pa}(S'_j))$. Globally factorizing the dynamics (Boutilier et al., 2000) and modeling each subprocess independently can lead to significant gains in sample efficiency (Kearns & Koller, 1999; Guestrin et al., 2003).

## 2.2 LEARNING FROM OFFLINE DATASETS

In this paper, we assume access to a dataset $\mathcal{D}$ with $K$ prerecorded agent experiences in the form of state-action trajectories $\tau_k = \{(s_0, a_0), ...., (s_{T_k}, a_{T_k})\}_{k=0}^{K-1}$ with continuous or discrete $s \in \mathcal{S}$ and actions $a \in \mathcal{A}$. The data can be collected using previously trained agents (to different levels of expertise) (Fu et al., 2020; Liu et al., 2023), through autonomous exploration (Hausman et al., 2018; Sharma et al., 2019; Sancaktar et al., 2022), via human teleoperation (Lynch et al., 2019; Schaal et al., 2005; Gupta et al., 2019) or a combination of those. Importantly, the data is potentially *unstructured* and does not necessarily contain trajectories that solve a specific downstream task, nor are they labeled with a reward signal (Urpí et al., 2023; Lynch et al., 2019; Pertsch et al., 2020). The goal of a learning algorithm is to leverage such a dataset to learn a goal-conditioned policy $\pi(a|s, g)$, where $g \in \mathcal{G}$ is used to condition the reward function $r(s, a, g)$. The optimal policy would then maximize the expected discounted sum of rewards $\mathbb{E}_\pi[\sum_{t=0}^{\infty} \gamma^t r(s_t, a_t, g)]$. For instance, the goal space can simply match the state space, and a sparse reward can be defined as an indicator function $r(s, a, g) = \mathbf{1}_{s=g}$. Depending on the employed downstream learning method, different requirements are imposed on the data. Our setting additionally requires that the action support is broad to be able to correctly estimate causal action influence.

# 3 METHOD

Autonomous agents should be able to act robustly under different environmental conditions. Accordingly, our method is designed to enable offline learning algorithms to learn a good policy in states that are not necessarily within the support of the data distribution. This is achieved by augmenting real data with counterfactual modifications to *causally action-unaffected* entities. We hypothesize that this will break spurious correlations that are otherwise picked up by policies and prevent them from generalizing. Crucially, we use a local causal graph formulation and rely on an independence assumption to explicitly compute causal influence through the Causal Action Influence (CAI) (Seitzer et al., 2021) measure. First, we introduce this formulation, the metric, and how it can be deployed to infer local causal connectivity. Subsequently, we will describe how this information can be used to produce counterfactual experience.

## 3.1 LOCAL CAUSAL GRAPHS

As elaborated in Section 2.1, the central components of a causal graph $\mathcal{G}$ are the transition kernels $P(S'_j \mid A, S) = P(S'_j \mid \mathrm{Pa}(S'_j))$ that describe the evolution of each entity $j$. Despite the assumptions

already made on the causal graph structure, in most non-trivial environments the graph is fully connected between timesteps: an edge between nodes $S_i/A$ and $S'_j$ is present as long as there is a single timestep for which $S_i/A$ affects $S'_j$ (Fig. 2(a)). Hence, the resulting factorized model does not bring any advantage over a simple monolithic representation.

In most environments, however, given a concrete timestep, in the majority of state configurations, there is limited interaction between entities and between entities and the agent. For example, given the state configuration in Fig. 2(b), the robot can only influence the kettle and its own end-effector through its actions, but none of the other entities.

With this in mind, we focus on the causal structure implied by a specific state configuration $S = s$. This is called the *local causal model* in $s$, as proposed in Seitzer et al. (2021); Pitis et al. (2020). The local causal graph with distribution $P_{\mathcal{V}}$ induced by observing $S = s$, has the joint distribution $P_{\mathcal{V}|S=s}$, which density factorizes as:

$$p(S' \mid A, S = s) = \prod_{j=1}^{N} p(S'_j \mid \mathrm{Pa}_{\mathcal{G}_s}(S'_j), S = s), \tag{2}$$

where the *state-conditioned local causal graph* $\mathcal{G}_{S=s}$ is the minimal factorization. In the local graph $\mathcal{G}_{S=s}$, the absence of an edge $(V, S'_j)$ for $V \in \{S_1, \ldots, S_N, A\}$ is implied by $S'_j \perp\!\!\!\perp V | S = s$, i.e. entity $j$'s next state is independent of $V$. An example is given in Fig. 2(b).

Given the present local perspective, to synthesize counterfactual experience, we are left with inferring the local factorization, i.e. discovering the conditional causal structure which is known to be a hard problem (Peters et al., 2017). Therefore, we make the key assumption that interactions between entities only rarely occur and are thus negligible. While the correctness of generated counterfactuals will rely on this assumption to hold, we argue that this is realistic in several robotics tasks of interest, including the ones we empirically evaluate. For example in the kitchen environment depicted in Fig. 1, the entities can hardly influence each other. In fact, the state of each entity is mostly controlled by the agent through its actions. This would also be the case in several manufacturing processes, in which interaction between entities should only happen under direct control of robots. Moreover, we remark that settings involving dense interaction between entities, and in which the assumption does not hold, remain a significant challenge for most heuristic methods for causal discovery Pitis et al. (2020), which would also underperform despite their generality. More formally, and in a graphical sense, we assume that there is no arrow $S_i \to S'_j, i \neq j$ as visualized by the gray dashed lines in Fig. 2(b). We note that only two groups of arrows remain in the causal graph: $S_j \to S'_j$, which we assume to always be present, and $A \to S'_j$.

Crucially, this practical assumption allows us to reduce the hard problem of local causal discovery to the more approachable problem of local action influence detection, that is, to predict whether given a specific state configuration, the agent can influence an entity through its actions in the next time step. As a result, instead of resorting to heuristics (Pitis et al., 2020), we can use an *explicit* measure of influence, namely the recently proposed method CAI (Seitzer et al., 2021), which we introduce below.

## 3.2 Causal Action Influence detection

To predict the existence of the edge $A \to S'_j$ in the local causal graph $\mathcal{G}_{S=s}$, Seitzer et al. (2021) use conditional mutual information (CMI) (Cover, 1999) as a measure of dependence, which is zero for independence. Therefore, in each state $S = s$ we use the point-wise CMI as a state-dependent quantity that measures causal action influence (CAI), given by

$$C^j(s) := I(S'_j; A \mid S = s) = \mathbb{E}_{a \sim \pi}\big[D_{KL}\big(P_{S'_j|s,a} \,\big|\big|\, P_{S'_j|s}\big)\big]. \tag{3}$$

The transition model $P_{S'_j|s,a}$ is modeled as a Gaussian neural network (predicting mean and variance) that is fitted to the training data $\mathcal{D}$ using negative log likelihood. The marginal distribution $P_{S'_j|s}$ is computed in practice using $M$ empirical action samples with the full model: $P_{S'_j|s} \approx \frac{1}{M} \sum_{m=1}^{M} P_{S'_j|s,a^{(m)}}, \ a^{(m)} \sim \pi$. We estimate the KL using an approximation for Gaussian mixtures from Durrieu et al. (2012). We refer the reader to Seitzer et al. (2021) for more details.

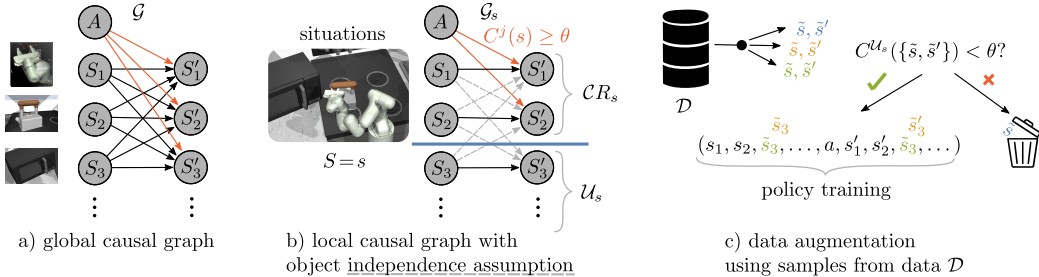

Figure 2: Illustration of counterfactual data augmentation. The global causal graph does not allow for factorization (a). Our local causal graph (b) is pruned by causal action influence. Object-object interactions are assumed to be rare/not existing (gray dashed). We swap elements not under control (in set $\mathcal{U}$) by samples from the data, thus creating alternative perceptions that yield the same outcome (c).

### 3.3 INFERRING LOCAL FACTORIZATION

Having introduced the concepts of locality and object independence, as well as a method to detect causal action influence, we proceed to infer the local factorization which will be leveraged to create counterfactual experience. For each state $s$ in our data set $\mathcal{D}$, we compute the uncontrollable set, as the set of entities in $s$ for which the agent has no causal action influence, expressed as:

$$\mathcal{U}_s = \{s_j \mid C^j(s) \leq \theta, j \in [1, N]\} \tag{4}$$

where $\theta$ is a fixed threshold. The set $\mathcal{U}_s$ contains all entities $j$ for which the arrow $A \rightarrow S'_j$ in the local causal graph $\mathcal{G}_s$ does not exist. The remaining entities are contained in the set of controllable entities $\mathcal{CR}_s = \{s_1, \ldots, s_N\} \setminus \mathcal{U}_s$. An illustration is given in Fig. 2(b).

With our assumptions and the sets $\mathcal{U}_s$ and $\mathcal{CR}_s$ we find that the local causal graph $\mathcal{G}_s$ is divided into the *disconnected* subgraphs $\mathcal{G}_s^{\mathcal{CR}}$, that contains the entities in $\mathcal{CR}$ and $A$, and into $|\mathcal{U}_s|$ *disconnected* subgraphs $\mathcal{G}_{s_i}^{\mathcal{U}}, i \in [1, |\mathcal{U}_s|]$, each of which contains an entity in $\mathcal{U}_s$ with only self-links, see Fig. 2(b). We can also compute the uncontrollable set for an extended time period, see Eq. 5 below.

### 3.4 COMPUTING COUNTERFACTUALS

Given the partitioning of the graph described above, we can think of each subgraph as an independent causal mechanism that can be reasoned about separately. Hence, we can create counterfactuals in the following way: given two transitions $(s, a, s')$ and $(\hat{s}, \hat{a}, \hat{s}') \in \mathcal{D}$ sampled for training, which have at least one uncontrollable subgraph structure in common (i.e. $\mathcal{U}_s \cap \mathcal{U}_{\hat{s}} \neq \emptyset$), we generate a counterfactual transition $(\tilde{s}, \tilde{a}, \tilde{s}')$ by swapping the entity transitions $(s_i, s'_i)$ with $(\hat{s}_i, \hat{s}'_i)$ and $i \in \mathcal{U}_s \cap \mathcal{U}_{\hat{s}}$.

However, in contrast to standard (global) causal considerations, local causal graphs introduce an additional constraint: the causal structure needs to stay the same after the intervention. Let us

---

**Algorithm 1: CAIAC**

**input** Dataset $\mathcal{D}$
    Compute uncontrollable set $\mathcal{U}_s, \forall s \in \mathcal{D}$ (Eq. 4).
    **while** `Training` **do**
        Sample $(s, a, s') \sim \mathcal{D}$
        $(\tilde{s}, \tilde{s}') \leftarrow (s, s')$
        **for** $s_i \in \mathcal{U}_s$ **do**
            Sample $(\hat{s}, \hat{a}, \hat{s}') \sim \mathcal{D}$
            **if** $\hat{s}_i \in \mathcal{U}_{\hat{s}}$ **then**
                $(\tilde{s}_i, \tilde{s}'_i) \leftarrow (\hat{s}_i, \hat{s}'_i)$
            **end if**
        **end for**
        Yield training samples $(s, a, s')$ and $(\tilde{s}, a, \tilde{s}')$
    **end while**

---

consider the counterfactual change above: the operation only strictly leaves the outcome of actions unchanged if $\mathcal{G}_s^{\mathcal{CR}} = \mathcal{G}_{\tilde{s}}^{\mathcal{CR}}$. The counterfactual $\tilde{s}$ should then be discarded if it alters the set of controllable entities: i.e. $\mathcal{CR}_s \neq \mathcal{CR}_{\tilde{s}}$. In practice, however, this operation is only possible when causal influence can be correctly measured in the counterfactual. As CAI, like previous heuristics, also relies on a learned transition model, the counterfactual is an out-of-distribution sample, and the output of the model will likely be inaccurate. In practice, we avoid this additional check and accept creating a small fraction of potentially unfeasible situations.

The pseudocode of our method, which we call **Ca**usal **I**nfluence **A**ware **C**ounterfactual Data Augmentation (CAIAC), is given in Algorithm 1.

## 4 RELATED WORK

**Data augmentation** Data augmentation is a fundamental technique for achieving improved sample-efficiency and generalization to new environments, especially in high-dimensional settings. In deep learning systems designed for computer vision, data augmentation can be found as early as in LeCun et al. (1998); Krizhevsky et al. (2012), which leverage simple geometric transformations, such as random flips and crops. Naturally, a plethora of augmentation techniques (Berthelot et al., 2019; Sohn et al., 2020) has been proposed over time. To improve generalization in RL, domain randomization (Tobin et al., 2017; Pinto et al., 2017) is often used to transfer policies from simulation to the real world by utilizing diverse simulated experiences. Cobbe et al. (2019); Lee et al. (2019) showed that simple augmentation techniques, such as cutout and random convolution, can be useful to improve generalization in RL from images. Similarly to us, (Laskin et al., 2020) use data augmentation for RL without any auxiliary loss. Crucially, most data augmentations techniques in the literature require human knowledge to augment the data according to domain-specific invariances (e.g., through cropping, rotation, or color jittering), and mostly target the learning from image settings. Nevertheless, heuristics for data augmentation can be formally justified through a causal invariance assumption with respect to certain perturbation on the inputs.

**Causal Reinforcement Learning** Detecting causal influence involves causal discovery, which can be pictured as finding the existence of arrows in a causal graph (Pearl, 2009). While it remains an unsolved task in its broadest sense, there are assumptions that permit discovery in some settings (Peters et al., 2012). Once the existence of an arrow can be detected, its impact needs to be established, for which several measures, such as transfer entropy or information flow, have been proposed (Schreiber, 2000; Lizier, 2012; Ay & Polani, 2008). In our case, we use conditional mutual information (Cover, 1999) as a measure of causal action influence, as proposed by Seitzer et al. (2021).

The intersection of RL and causality has recently been studied to improve interpretability, sample efficiency, and to learn better representations (Buesing et al., 2018; Bareinboim et al., 2015; Lu et al., 2018; Rezende et al., 2020). In particular, our work is related to that of Pitis et al. (2020), which also leverages influence detection to generate counterfactual data. However, they aim at estimating the entire local causal graph, which is a challenging problem. In practice, they rely on a heuristic method based on the attention weights of a transformer world model which does not scale well to high-dimensional environments. In contrast, our method does not require learning the entire local causal graph, as it assumes that the interactions between entities (except the agent) are sparse enough to be neglected. We remark that this is a reasonable assumption in many robotic experiments, such as the ones we are considering. This also implies that the agent is the only entity that can influence the rest of the entities through its actions. Therefore, this setting is related to the concept of contingency awareness from psychology (Watson, 1966), which was interestingly already considered in deep reinforcement learning methods for Atari (Song et al., 2020; Choi et al., 2018).

## 5 EXPERIMENTS

We evaluate CAIAC in two goal-conditioned settings: offline RL and offline self-supervised skill learning. In particular, we are interested in evaluating whether CAIAC

1. leads to better generalization to unseen configurations,

2. enlarges the support of the joint distribution over the state space in low data regimes, and

3. works as an independent module combinable with any learning algorithm of choice (in particular, offline RL and skill-based behavioral cloning).

**Baselines** We compare CAIAC with CODA (Pitis et al., 2020), a counterfactual data augmentation method, which uses the attention weights of a transformer model to estimate the local causal structure. Given two transitions that share local causal structures, it swaps the connected components to form new transitions. Additionally, we compare with an ablated version of CODA, CODA-ACTION, which only estimates the influences of the action using the transformer weights and thus is a 'heuristic'-sibling of our method. As an ablation, we include a baseline without data augmentation (NO-AUGM). Extended results on the impact of the ratio of observed and counterfactual data are in Appendix A.3.

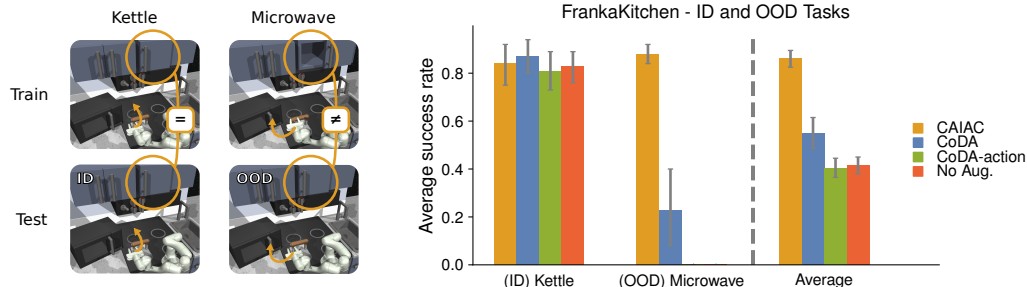

Figure 3: Motivating Franka-Kitchen example. The experimental setup (left) and success rates for in-distribution and out-of-distribution tasks (right). Metrics are averaged over 10 seeds and 10 episodes per task, with 95% simple bootstrap confidence intervals.

### 5.1 GOAL CONDITIONED OFFLINE SELF-SUPERVISED SKILL LEARNING

Our initial experiments investigate whether CAIAC can increase the generalization capabilities of algorithms when learning from demonstrations. We apply our method to the challenging Franka-Kitchen environment from Gupta et al. (2019). We make use of the data provided in the D4RL benchmark (Fu et al., 2020), which consists of a series of teleoperated sequences in which a 7-DoF robot arm manipulates different parts of the environment (e.g., it opens microwave, switches on the stove). Crucially, all demonstrations are limited to a few manipulation sequences (for example, first opening the microwave, turning on a burner, and finally the light). Thus, the support of the joint distribution over entities in the environment is reduced to only a few combinations. To illustrate this using a concrete example, the light is never on if the microwave hasn't been opened yet. When dealing with human demonstrations in large environments, this type of selective data sampling is to be expected, since the number of possible combinations explodes with the number of entities in the environment. Hence, given the limited human labor time, only a few combinations will be included in the data.

We hypothesize that CAIAC will create valid counterfactual data such that the downstream learning algorithms would be able to generalize to unseen state configurations. As a downstream learning algorithm we use LMP (Lynch et al., 2019), an offline goal-conditioned self-supervised learning algorithm, which learns to map similar behaviors (or state-action trajectories) into a latent space from which goal-conditioned plans can be sampled. Formally, LMP is a sequence-to-sequence VAE (Sohn et al., 2015; Bowman et al., 2015) autoencoding random experiences extracted from the dataset through a latent space. In our case, we use experiences of fixed window length $\kappa$. Given the inherent temporal abstraction of the algorithm, we generate counterfactuals of fixed length $\kappa > 1$ by computing the uncontrollable set $\mathcal{U}_{(s_t, s_{t+1}, \ldots, s_{t+\kappa})}$ for the entire window as the intersection over all time slices, where

$$\mathcal{U}_{(s_t, s_{t+1}, \ldots, s_{t+\kappa})} = \bigcap_{\tau=t}^{t+\kappa-1} \mathcal{U}_{s_\tau}. \tag{5}$$

For specific details on the learning algorithm and the Franka-Kitchen environment, we refer to A.1.1 and A.2.1 respectively.

### 5.1.1 FRANKA-KITCHEN: MOTIVATING EXPERIMENT

Our first experiment is designed to verify claim 1), i.e., that CAIAC enables generalization to unseen configurations over entities. First, we showcase this in a simple and controlled environment. Thus, we create a reduced modified dataset from the original D4RL dataset (Fu et al., 2020), that contains only demonstrations for the microwave task (MW) and the kettle (K) task. During demonstrations for the (MW) task, we initialize the cabinet to be always open, whereas for demonstrations for the (K) task, it remains closed. The rest of the objects are set to the default initial configuration (see A.2.1). At inference time, we initialize the environment with its default initial configuration (crucially, the cabinet is closed), and we evaluate both tasks ((K) and (MW)), as shown in Fig. 3(left). Hence, while the (K) task was demonstrated for the current configuration (in-distribution, ID), the agent is effectively evaluated on an out-of-distribtion (OOD) configuration for the (MW) task.

Table 1: Average success rates for Franka-Kitchen tasks with OOD initial configurations, computed over 10 seeds and 10 episodes per task with 90% simple bootstrap confidence intervals.

| Algorithm | CAIAC | CoDA | CoDA-action | No-Augmentation |
|---|---|---|---|---|
| Kettle | **0.41 ± 0.06** | 0.18 ± 0.05 | 0.16 ± 0.1 | 0.06 ± 0.04 |
| Microwave | **0.30 ± 0.06** | 0.07 ± 0.05 | 0.0 ± 0.03 | 0.01 ± 0.03 |
| Bottom-burner | **0.10 ± 0.07** | 0.01 ± 0.01 | 0.0 ± 0.0 | 0.0 ± 0.0 |
| Slide cabinet | 0.04 ± 0.01 | **0.10 ± 0.05** | 0.02 ± 0.02 | 0.06 ± 0.03 |
| Light switch | **0.03 ± 0.03** | 0.0 ± 0.0 | 0.0 ± 0.0 | **0.00 ± 0.04** |
| Hinge cabinet | 0.0 ± 0.0 | 0.0 ± 0.0 | 0.0 ± 0.0 | 0.0 ± 0.0 |

We evaluate success rate on both tasks with CAIAC and all baselines, as shown in Fig. 3(right). All methods are able to solve the (K) task, as expected, since it is in-distribution (ID), and can be solved by simple goal-conditioned behavioral cloning. However, we observe fundamentally different results for the OOD (MW) task. In principle, CAIAC can detect that the sliding cabinet is never under control of the agent, and will be able to create the relevant counterfactuals. Indeed, the performance of CAIAC in the OOD task (MW) is not affected, and it is the same as for the ID task. On the other hand, the performance of CoDA and CoDA-ACTION is drastically impaired in the OOD setting. Despite the simplicity of the setting, the input dimensionality of the problem is high, and the transformer attention weights are not able to recover the correct causal graph. By picking up on spurious correlations, the attention weights of the transformer estimate low influence from the action to all entities (even the agent), and hence CoDA-ACTION creates dynamically-unfeasible counterfactuals which affect performance. Since the ratio of observed-counterfactuals data is 1:1 we hypothesize that there is enough in-distribution data to not affect the (K) task for CoDA-ACTION. The local graph induced by CoDA has at least as many edges as the one of CoDA-ACTION, and hence the probability for creating unfeasible counterfactuals is lower. We hypothesize, that despite not learning correct causal influence, it might still provide some samples which benefit the learning algorithm and allow for an average OOD success rate of 0.2. We refer the reader to Appendix A.3 for further analysis on the impact of the ratio of observed:counterfactual data for this experiment. Finally, as expected, No AUGM. fails to solve the OOD (MW) task.

### 5.1.2 FRANKA-KITCHEN: ALL TASKS

Having evaluated CAIAC in a controlled setting, we now scale up the problem to the entire Franka-Kitchen D4RL dataset. While in the standard benchmark the agent is required to execute a single fixed sequence of tasks, we train a goal-conditioned agent and evaluate on the full range of tasks, which include the microwave, the kettle, the slider, the hinge cabinet, the light switch and the bottom left burner tasks Mendonca et al. (2021). One task is sampled for each evaluation episode. While alleviating the need for long-horizon planning, this results in a challenging setting, as only a subset of tasks is shown directly from the initial configuration. However, the largest challenge in our evaluation protocol lies in the creation of unobserved state configurations at inference time. While the provided demonstrations always start from the same configuration (e.g., the microwave is always initialized as closed), at inference time, we initialize all non-target entities (with $p = 0.5$) to a random state, hence exposing the agent to OOD states. We expect that agents trained with CAIAC will show improved performance to unseen environment configurations, as those can be synthesized through counterfactual data augmentation. The results, shown in Table 1, are consistent with the challenging nature of this benchmark, as the evaluated tasks involve OOD settings in terms of states and actions. Nevertheless, we find that CAIAC is significantly better than baselines in 4/7 tasks, while being on par with the best method in the remaining 3. We hypothesize that the low performance on these 3 tasks is due to the absence of robot state and action trajectories in the dataset that show how to solve each of the 3 tasks from the initial robot joint configuration. Hence, even with perfect counterfactual data augmentation these tasks remain challenging. We refer the reader to the Appendix A.2.1 for further analysis. As observed in the simplified setting, methods relying on heuristic-based causal discovery (CoDA and CoDA-ACTION) suffer from misestimation of causal influence, and thus from the creation of dynamically-unfeasible training samples. Without any data augmentation, the learning algorithm cannot perform the OOD tasks. We refer the reader to Fig. 6 for a visualization of the computed CAI scores per each entity on one of the demonstrations for the Franka-Kitchen dataset.

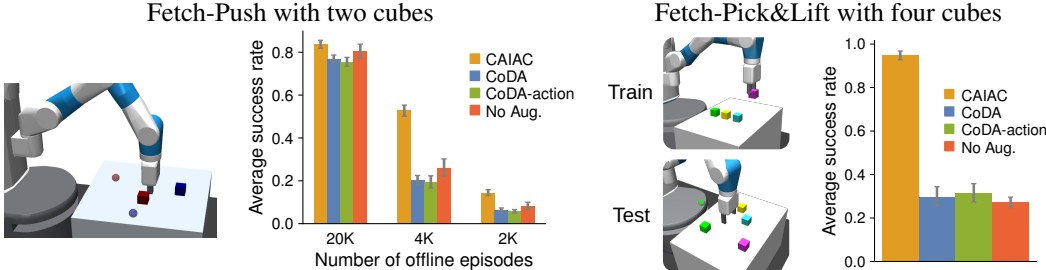

Figure 4: Success rates for Fetch-Push with 2 cubes (left) and Fetch-Pick&Lift with 4 objects (right). Metrics are averaged over 30 seeds and 50 episodes with 95% simple bootstrap confidence intervals. Fetch-Push investigates scarce data regimes; Fetch-Pick&Lift has abundant data (40k episodes).

## 5.2 Goal-conditioned Offline RL: Robotic environments with 2 and 4 cubes

With the following experiments, our aim is to verify claim 2), i.e., that CAIAC can enlarge the support of the joint distribution in low data regimes. We first evaluate CAIAC in Fetch-Push, where a robotic arm has to slide two blocks to target locations. For this experiment we collect 20k trajectories using an expert policy (30%) and random policy (70%) and train an agent offline using TD3 (Fujimoto et al., 2018) in different data regimes: namely 100% of data, 20%, and 10%. We use HER (Andrychowicz et al., 2017) to relabel goals with the `future` strategy on real data and with the `random` strategy on counterfactual data. More details are given in Appendices A.2.2, A.1.2, and A.5. We compare success rates between baseline and CAIAC among different data regimes in Fig. 4 (left).

For high data regimes, CAIAC and No Augm. baseline perform similarly given that there is enough coverage of the state space in the original dataset. In the medium data regime there is a significant performance gain. Given sufficient support for the marginal distribution on the state of each entity, CAIAC can substantially increase the support of the joint distribution, leading to higher performance. In the smallest data regime, we see that CAIAC is still significantly better than the baselines, but the performance gain is reduced, likely due to the lack of marginal distribution coverage for each entity. Transformer-based methods CoDA and CoDA-action seem to create detrimental counterfactuals in all data regimes leading to decreased performance. The estimated influence scores for all the methods are visualized in A.4. Details on threshold optimization are provided in Appendix A.5. We note that, while previous work (Pitis et al., 2020) has shown good online performance of CoDA in this environment, it resorted to a handcrafted heuristic to decide about influence.

We also test CAIAC on Fetch-Pick&Lift, a modified version of the Fetch-Pick&Place environment with 4 cubes (Fig. 4 (right)). At training time, the blocks are aligned and the robot needs to lift a desired cube. At test time, the cubes are randomly arranged on the table. We use HER with the `future` strategy (for counterfactual samples, trajectories are augmented before goals sampling). Results in an abundant data regime (40k episodes) are shown in Fig. 4 (right). Even in this high data regime, when there is a mismatch between joint state distributions at training time and test time, CAIAC shows drastic performance improvements over all baselines.

This setting also confirms claim 3) that CAIAC can be applied to different methods: hierarchical behavioral cloning and flat reinforcement learning, trained on near-expert and mostly random data.

## 6 Discussion

While extracting complex behaviors from pre-collected datasets is a promising direction for robotics, data scarcity remains a principal issue in high-dimensional, multi-object settings, due to a combinatorial explosion of possible state configurations which cannot be covered densely by demonstrations. Hence, current learning methods often pick up on spurious correlations and struggle to generalize to unseen configurations. In this paper, we proposed CAIAC as a method for counterfactual data augmentation without the need for additional environment interaction nor counterfactual model rollouts, which can be used with any learning algorithm. By adding an inductive bias on the causal structure of the graph, we circumvented the problem of full causal discovery and reduced it to the computation of an explicit measure of the agent's causal action influence over objects. Empirically, we show that CAIAC leads to enhanced performance and generalization to unseen configurations, suggesting that further advances in addressing both partial and full causal discovery problems can be substantially beneficial for robot learning.

**Reproducibility Statement**   In order to ensure reproducibility of our results, we make our code-base publicly available at https://sites.google.com/view/caiac, and provide detailed instructions for training and evaluating the proposed method. Furthermore, we describe algorithms and implementation details in Appendix A. Finally, as our experiments rely on offline datasets, we publish them at the same link.

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

# A  APPENDIX

## A.1  IMPLEMENTATION OF DOWNSTREAM LEARNING ALGORITHMS

In this section, we report implementation details concerning the learning algorithms. For a fine-grained description of all hyperparameters, we refer to our codebase at `https://sites.google.com/view/caiac`.

### A.1.1  GOAL-CONDITIONED OFFLINE SELF-SUPERVISED SKILL LEARNING

For the goal-conditioned self-supervised learning experiments we used LMP (Lynch et al., 2019), a goal-conditioned self-supervised method. It consists of a stochastic sequence encoder, or learned posterior, which maps a sequence $\tau$ to a distribution in latent plan space $q(z|\tau)$, a stochastic encoder or learned goal-conditioned prior $p(z|s, g)$ and a decoder or plan and goal conditioned policy: $\pi(a|z, s, g)$. During training, the self-supervised goals $g$ for counterfactual samples are relabeled by sampling the final state of the skill *after* augmenting it. The main difference with the original implementation is that the latent goal representation is only added to the prior, but not the decoder. Additionally, we also implemented KL balancing in the loss term between the learned prior and the posterior: we minimize the KL-loss faster with respect to the prior than the posterior. Given that the KL-loss is bidirectional, in the beginning of training, we want to avoid regularizing the plans generated by the posterior towards a poorly trained prior. Hence, we use different learning rates, $\alpha = 0.8$ for the prior and $1 - \alpha$ for the posterior, similar to Hafner et al. (2020). These two modifications were also suggested in (Rosete-Beas et al., 2022). Additionally, our decoder was open-loop (instead of close loop): given a sampled latent plan $z$ it decodes the whole trajectory of length `skill length`$= N$, i.e. our decoder is $\pi(\hat{a}|z)$, where $\hat{a} = a_t, ..., a_{t+N}$ is the sequence of decoded actions, instead of $\pi(a|z, s, g)$. This modification was needed due to the *skewness* of the dataset. Since the demonstrations were provided from an expert agent, given most of the states, the distribution over actions is unimodal: when the robot is close to the microwave, the only sequence of actions in the dataset is the one that opens the microwave. Hence, a close-loop decoder would learn to ignore the latent plan, and only rely on the state. To solve this issue, we make the decoder open-loop.

### A.1.2  GOAL-CONDITIONED OFFLINE RL

For the goal-conditioned offline RL setting we implement the TD3 algorithm (Fujimoto et al., 2018) with HER (Andrychowicz et al., 2017). Unless specified differently, the hyperparameters used were the ones from the original implementation. We use HER (Andrychowicz et al., 2017) to relabel the goals for real data, with a `future` relabeling strategy with $p = 0.5$, where the time points were sampled from a geometric distribution with $p_{geom} = 0.2$. For the counterfactual data we relabel the goals with $p = 0.5$ random sampling from the achieved goals in the buffer of counterfactual samples. In the experiments, we realized that the relabeling strategy had an impact on the performance of the downstream agent. To disentangle the impact of the relabeling strategy from the impact of the counterfactual data generation and to ensure a fair comparison, we also relabeled the same percentage of goals (i.e. $p = 0.25$) with `random` strategy for the No Augm. baseline. We train each method for 1.2M gradient steps, although all methods reach convergence after 600k gradient steps. The percentage of counterfactuals in each batch is set to $0.5$.

## A.2  EXPERIMENTAL DETAILS

### A.2.1  FRANKA-KITCHEN

We use the kitchen environment from the D4RL benchmark (Fu et al., 2020) which was originally published by Gupta et al. (2019). The D4RL dataset contains different dataset versions: `kitchen-complete`, `kitchen-partial`, `kitchen-mixed`, which contain 3690, 136950 and 136950 samples respectively, making up to approximately 14 demonstrations for `kitchen-complete` and 400 demonstrations for each `kitchen-partial` and `kitchen-mixed`. The simulation starts with all of the joint position actuators of the Franka robot set to zero. The doors of the microwave and cabinets are closed, the burners turned off, and the light switch also off. The kettle will be placed in the bottom left burner. The observation are 51-dimensional, containing the joint positions of the robot (9 dim), the positions of the all the kitchen

items (21 dim) and the goal positions of all the items (21 dim). The length of the episode is 280 steps, but the episode will finish earlier if the task is completed. The task is only considered solved when all the objects are within a norm threshold of 0.3 with respect to the goal configuration. While in the standard benchmark the agent is required to execute a single fixed sequence of tasks, we train a goal-conditioned agent, and evaluate on one task per each evaluation episode. For the Franka-Kitchen motivating example (see 5.1.1) we query for either the kettle or the microwave task, in the Franka-Kitchen: All tasks (see 5.1.2) we query for the full range of tasks, which include the microwave, the kettle, the slider, the hinge cabinet, the light switch and the bottom left burner tasks. While alleviating the need for long-horizon planning, this results in a challenging setting, as only a subset of tasks is shown directly from the initial configuration. Specifically out of the 1200 demonstrations in the dataset, containing different task sequences, only 3 objects are shown to be manipulated from the initial robot configuration: 60% of the trajectories solve the microwave task first, 30% show the kettle task first and 10% show the bottom burner first. This aligns with the relative performance achieved for those tasks. For the 3 remaining tasks, namely the slide cabinet, the light and the hinge cabinet, there is no demonstration shown directly from the initial configuration and hence the low performance.

**Franka-Kitchen: motivating experiment** For the first experiment (see Subsection 5.1.1), we modify the dataset version `kitchen-mixed` to only contain $\sim 50$ demonstrations of length $\sim 40$ timesteps for each `(mw)` and `(k)` task. During demonstrations for the `(mw)` task, we initialize the cabinet to be always open, whereas for demonstrations for the `(k)` task, it remains closed. The rest of the objects are set to the default initial configuration. The goal configuration for all the objects was set to their initial configuration (as defined above), except for the microwave or the kettle, which were set to the default goal configuration when querying for the `(mw)` and `(k)` tasks respectively.

**Franka-Kitchen: all tasks** For the second experiment (see Subsection 5.1.2) we merge the 3 provided datasets `kitchen-complete`, `kitchen-partial`, `kitchen-mixed`. For this experiment, each object (except the one related to the task at hand to ensure non-trivial completion), was randomly initialized with $p = 0.5$, otherwise it was initialized to the default initial configuration (as defined above). We then modify the desired goal to match the initial configuration for all non-target entities.

### A.2.2 FETCH-PUSH WITH 2 CUBES

Expert data for the experiment in Subsection 5.2 includes 6000 episodes collected by an agent trained online using TD3 and HER up to approximately 95% success rate. We additionally collect 14000 episodes with a random agent, which make up for the random dataset. This sums up to a total of 20000 episodes (each of length 100 timesteps), with 30% expert data and 70% random data. Initial positions and goal positions of the cubes are sampled randomly on the table, whereas the robot is initialized in the center of the table with some additional initial random noise . The rewards are sparse, giving a reward of $-1$ for all timesteps, except a reward of $0$ when the position of each of the 2 blocks are within a $2-$norm threshold of $0.05$. The observation space is 34-dimensional, containing the position and velocity of the end effector (6dim), of the gripper (4dim) and the object pose, linear and rotational velocities of the objects (12dim each). In contrast to the original `Fetch-Push-v1` (Plappert et al., 2018) environment and similarly to Pitis et al. (2020) we do include parts of the state space accounting for relative position or velocities of the object with respect to the gripper, which would entangle the two. The goal is 6-dimensional encoding the position for each of the objects. The action space is 4-dimensional encoding for the end-effector position and griper state. At test time we count the episode as successful upon reaching the goal configuration (i.e., observing a non-negative reward).

### A.3 ABLATION: RATIO OF OBSERVED-TO-COUNTERFACTUAL DATA

In this section, we study the effect of the ratio of observed-to-counterfactual data generated with CAIAC, by evaluating downstream performance on the Franka-Kitchen motivating example, as presented in 5.1.1. Empirical results for this ablation are shown in Fig. 5. As expected, we observe that the ratio of counterfactuals does not have any significant impact on the success rate on the `(k)` task. This is because the task is evaluated in distribution, and hence the downstream learning algorithm does not require observing counterfactual experience (but still doesn't suffer from it). For the OOD `(mw)` task we see that increasing the number of counterfactuals up to a 0.9 ratio has a positive effect in performance, leading the agent to generalize better to the OOD distribution.

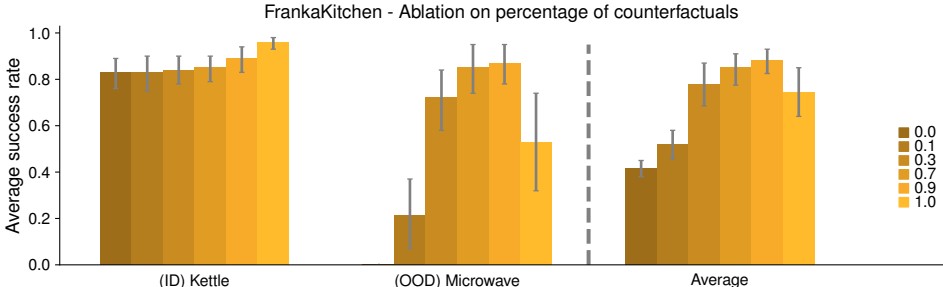

Figure 5: Performance of CAIAC on motivating Franka-Kitchen example when controlling the percentage of counterfactual samples in each batch. Metrics are averaged over 10 seeds and 10 episodes per task, with 95% simple bootstrap confidence intervals.

However, when the ratio is increased up to 1, we only use synthesized counterfactual data. We observe a decrease in performance with high variance among training seeds. This hard ablation shows the need of real data during training to avoid induced selection bias, as also observed in (Pitis et al., 2020). With a ratio 0.0, we recover the performance of the No Augm. baseline.

### A.4 Details on Influence Detection Evaluation

To detect causal action influence we use CAI, as described in 3.2.

$$C^j(s) := I(S'_j; A \mid S = s) = \mathbb{E}_{a \sim \pi}\big[D_{KL}\big(P_{S'_j|s,a} \,\big\|\, P_{S'_j|s}\big)\big]. \tag{6}$$

For it we need to learn the transition model $P_{S'_j|s,a}$.

As an example, in the case of robotic manipulation environments *physical contact* is not needed for causal action influence as long as the agent can change the object pose, even if indirectly, in a single simulation step.

**World model training** For the Franka-Kitchen experiments all models were trained to predict the full state of the environment. For increased performance in the Fetch-Push task, all models were trained to predict the next position of the end effector of the agent gripper and of the objects (3 dimensions each). For CAIAC the transition model $P_{S'_j|s,a}$ is modeled as a Gaussian neural network (predicting mean and variance) that is fitted to the training data $\mathcal{D}$ using negative log likelihood. We used a simple multi-layer perceptron (MLP) with two separate output layers for mean and variance. To constrain the variance to positive range, the variance output of the MLP is processed by a softplus function (given by $\log(1 + exp(x))$), and a small positive constant of $10^{-8}$ was added to prevent instabilities near zero. We also clip the variance to a maximum value of 200. For weight initialization, orthogonal initialization is used. For the Franka-Kitchen we use larger MLPS, with 3 layers for the simplified and 4 layers for the full experiment, each with 256 units and a learning rate of $8e^{-4}$.

For `CODA` and `CODA-action` we use a self-implementation of the transformer model. We use a model with 3 layers and 4 attention heads for the Fetch-Push task and 5 layers and 4 heads for all the Franka-Kitchen tasks, with an embedding space and output space of 128 dimensions each. We also used a learning rate of $8e^{-4}$.

All models were trained for 100k gradient steps, and tested to reach low MSE error for the predictions in the validation set (train-validation split of 0.9-0.1). We trained all models using the Adam optimizer Kingma & Ba (2014) , with default hyperparameters.

In general, the models were trained using the same data as for the downstream task for all experiments. However, for the Franka-Kitchen task, we add some additional collected data on the environment when acting with random actions. The reason is that, in order to compute CAI, we query the model on randomly sampled actions from the action space. Due to the expert nature of the kitchen dataset comes from an informed agent, the original dataset might lack random samples and hence we would query the model OOD when computing CAI. For both experiments in the Franka-Kitchen simplified experiment we added 1x the original dataset of random data. Further experiments on the impact of the amount of random data could be beneficial. This was not needed for the Fetch-Push task since the

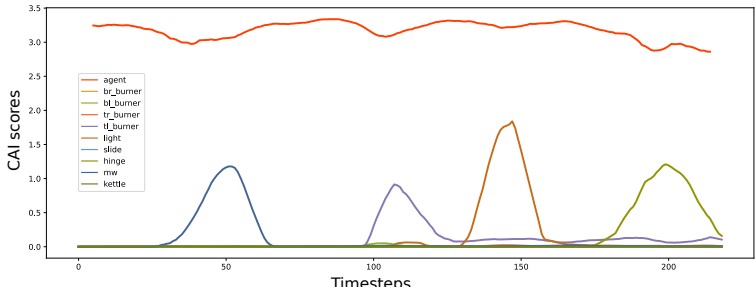

Figure 6: Computed CAI scores per each entity on one of the demonstrations for the Franka-Kitchen dataset. We can observe how the influence of the agent's actions over objects changes over time. First influencing the microwave (mw), then the top-left bottom burner (tl_burner), then the light switch (light) and finally the hinge cabinet (hinge). We selected a threshold of $\theta = 0.3$.

dataset already contains random action. We note that this additional data is also provided for training all transformer-based baselines.

**CAI scores** In practice, we compute the cai scores using the estimator:

$$C^j(s) = \frac{1}{K} \sum_{i=1}^{K} [D_{KL}\big(p(s'_j \mid s, a^{(i)}||\frac{1}{K} \sum_{k=1}^{K} p(s'_j \mid s, a^{(k)})\big)] \tag{7}$$

with $a \sim \pi$, where $\pi(A) := \mathcal{U}(\mathcal{A})$ (i.e. a uniform distribution over the action space) and with K = 64 actions. We refer the reader to (Seitzer et al., 2021) for more details. In Fig. 6 we show the computed CAI scores over a demonstration in the Franka-kitchen and in Fig. 7 (top) over an episode for the Fetch-Push task. Given the scores we need a threshold $\theta$ to get a classification of control (see Equation 4). For Fetch-Push we set $\theta = 0.05$, for Fetch-Pick&Lift we set $\theta = 2.0$ and for Franka-Kitchen $\theta = 0.3$ . See section A.5 for a thorough analysis on the impact of the influence threshold $\theta$ and how the parameter was chosen for each of the methods.

**Transformer scores** To compute causal influence, we use the attention weights of a Transformer, where the score is computed as follows. Letting $A_i$ denote the attention matrix of the i'th of N layers, the total attention matrix is computed as $\prod_{i=1}^{N} A_i$. For CoDA the score is computed by checking the corresponding row $i$ and column $j$ for the check $s_i \rightarrow s'_j$, whereas for CoDA-action we restrict ourselves to the row corresponding to the input position of the action component, and the output position of the object component. Our implementation follows Pitis et al. (2020), to which we refer for more details. In Fig. 7 (bottom) and Fig. 8, we show respectively the computed CoDA-action and CoDA scores over an episode for the Fetch-Push task. Given the scores we need a threshold $\theta$ to get a classification of control. For Fetch-Push we set $\theta = 0.2$ for CODA and CODA-action, for Fetch-Pick&Lift we set $\theta = 0.2$ for CODA-action and $\theta = 0.15$ for CODA. For Franka kitchen we set $\theta = 0.3$ for all methods. See section A.5 for a thorough analysis on the impact of the influence threshold $\theta$ and how the parameter was chosen for each of the methods.

## A.5 ANALYSIS ON INFLUENCE THRESHOLD

To get a classification of control, we optimize the value for the threshold $\theta$ for all methods. We train 10 different world models for each method and we run a grid search over the parameter $\theta$. We run 3 seeds for each of the 10 models and we picked the value for $\theta$ that optimizes the downstream task average performance among the 10 models and the 3 seeds. In Figure Fig. 9 we plot the ROC curves for correct action influence detection for both CAIAC and CoDA for the Fetch-Push task. In such an environment, one can specify a heuristic of influence using domain knowledge, namely the agent doesn't have influence on the object if 7cm apart. An accurate model generates an Area Under the Curve (AUC) close to 1, while a random model stays along the diagonal. In Fig. 9 we can observe that the attention weights of the transformer world model aren't accurate for detecting influence. Additionally, there is a high variability on the different trained transformers (see also Fig. 11) , making it hard to optimize for the threshold $\theta$ for this type of model architecture. In contrast, we see that ROC curves for CAI have an AUC $\approx 0.9$ and hence it is an accurate measure for predicting influence

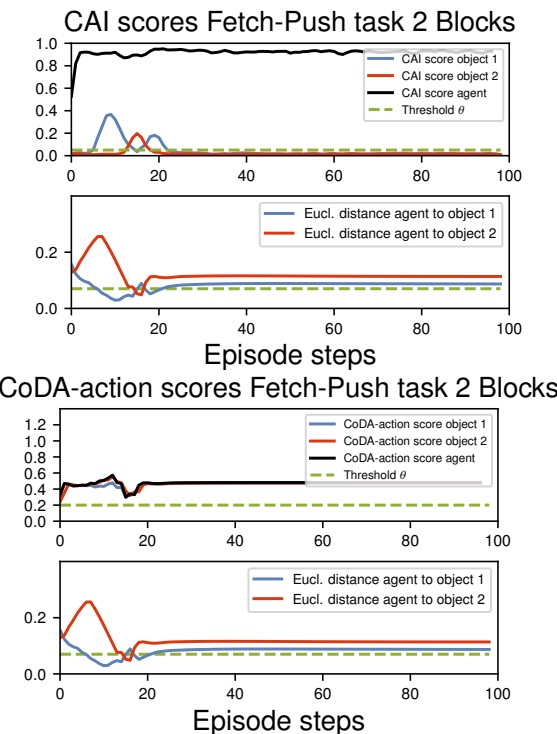

Figure 7: (1/4) from Top: Computed CAI scores per object on one of the expert demonstrations for the Fetch-Push dataset. We can observe how the influence of the agent's actions over objects changes over time. First pushing object 1, then object 2 and object 1 again. In green we show the optimized threshold $\theta = 0.05$ for this task. (2/4): We show distance from the robot end-effector to each of the objects as a domain knowledge heuristic for action influence. In green we show the heuristic distance of 7cm that we use as a threshold to consider the agent can influence the object within the next timestep. (3/4). Computed CoDA-action scores on the same episode as above. In green we show the optimized threshold $\theta = 0.2$ for this task.

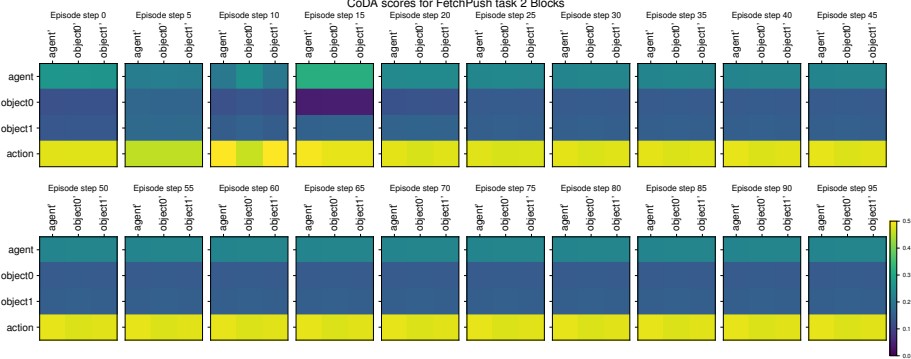

Figure 8: Computed CoDA scores on the same episode as in Fig. 7. We show snapshots of the attention weights of the transformer every 5 time steps as computed by the CoDA algorithm. The element $i, j$ in the matrix shows the attention (or influence) $s_i \rightarrow s'_j$. We see how the transformer completely fails on discovering the full causal graph, being even unable to recover influence along the diagonal, i.e., that an entity state at time t influences the entity state at time t+1. We made sure the models are trained properly and achieve good predictive performance.

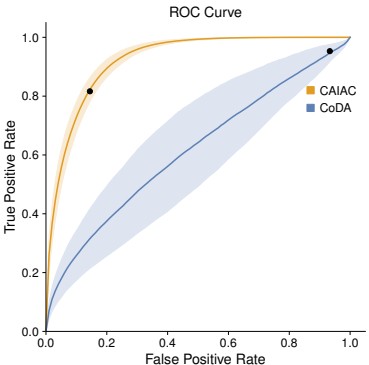

Figure 9: ROC curves for CAIAC and `CoDA-action` (averaged over 10 trained world models and 1 standard deviation shaded). We show measures true and false positive rates (TPR and FPR) while sweeping the influence threshold $\theta$. In black we show the corresponding TPR and FPR for the optimal $\theta$ for both methods. See also Figs. 10, 11.

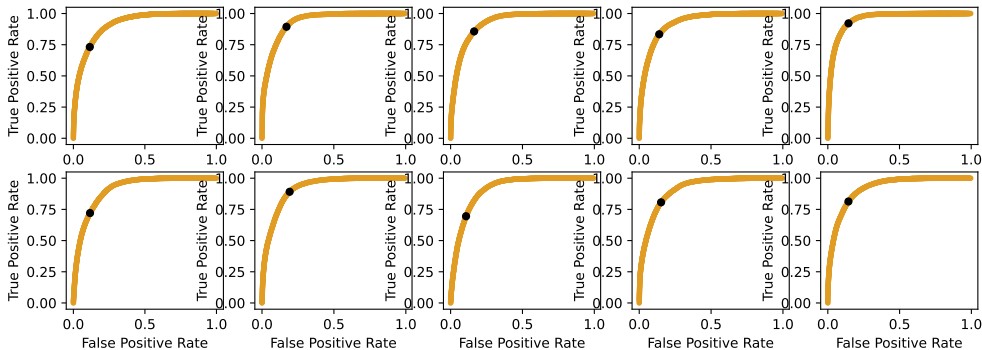

Figure 10: ROC curves for all the 10 trained world using CAIAC. In black we show the corresponding true postive and false positive rate for the optimal threshold $\theta = 0.05$.

in the Fetch-Push environment. Additionally, given its low variance across training seeds (see also Fig. 10), same thresholds reach the same TPR/FPR across models, making it easy to optimize for $\theta$.

## A.6 COMPUTATIONAL DEMANDS

CAIAC relies on computing the CAI measure for data augmentation. In turn, CAI can be evaluated for all entities at once, through $k$ forward passes for the $k$ counterfactual actions, which are performed in a batch-wise fashion. $k$ is a constant factor, and does not scale with the number of entities. Methods relying on a transformer world model, like CoDA and CoDA-action only need one forward pass (which internally has quadratic cost in the number of entities due to cross attention). However, CoDA also needs to compute the connected components from the adjacency matrix, which has a quadratic cost. For relatively few entities, as is common in the robotic manipulation environments, the computational overhead is relatively small. For the high data regime Fetch-Push environment we timed how long it takes for each algorithm to compute influence on all 2M datapoints and provide the results in Table 2. The algorithms were benchmarked on a 12-core Intel i7 CPU. Additionally, counterfactuals could be generated in parallel to the learning algorithm and hence not significantly impact runtime of the algorithm. Furthermore, in our offline setting, counterfactuals can be fully precomputed.

## A.7 ANALYSIS ON THE QUALITY OF CREATED COUNTERFACTUALS

We provide an analysis on the created counterfactuals using CAIAC. We visually show how the created augmented samples have an increased support of the joint state space distribution in the

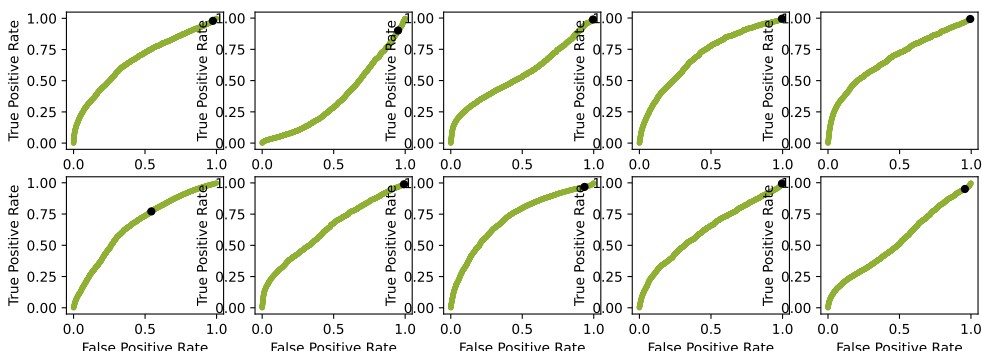

Figure 11: ROC curves for all the 10 trained world using `CoDA-action`. In black we show the corresponding true postive and false positive rate for the optimal threshold $\theta = 0.2$.

kitchen dataset, compared to the No-Augmented baseline. In Fig. 12 we show 1000 randomly selected samples from the Franka-Kitchen dataset together with one counterfactual for each. The visualization employs t-SNE for a 2D representation of the high-dimensional state space. We observe how the augmented samples using CAIAC cover a much larger space than the observed data, suggesting that we are able to provide states with a richer amount of configurations. However, this does not tell us whether these counterfactuals are actually valid. To quantify whether an augmented datapoint is valid, we do the following: We consider time-windows $(s_t, \ldots, s_{t+\tau})$ sampled from the Franka-Kitchen dataset. We create counterfactuals for this window with CAIAC and `CoDA`, $(\tilde{s}_t, \ldots, \tilde{s}_{t+\tau})$. We reset the simulator to $\tilde{s}_t$ and simulate the future with the action sequence in the original trajectory for $\tau$ steps. If the counterfactual is valid, then resulting state of the simulation should coincide with the counterfactual $\tilde{s}_{t+\tau}$. Unfortunately, the simulator is not perfectly deterministic (or we are unable to set the full simulator state), so measuring exact matching states is not possible. Thus, we run the simulator $K = 10$ times with different env-seeds and obtain $\mathcal{S}' = \{\bar{s}_{t+\tau}^k\}_{k=1}^K$. We fit a multivariate Gaussian to the set $\mathcal{S}'$. We now compute the probability of the counterfactual $\tilde{s}_{t+\tau}$ under the Gaussian distribution. High probability means valid counterfacturals, low means invalid. In Fig. 13 we present the histogram of log probabilities. The counterfactuals by CAIAC are mostly valid, which is not the case for the `CoDA` baselines.

Table 2: Computational demands for computing counterfactuals for the different algorithms. Runtime was benchmarked on a 12-core Intel i7 CPU.

|  | CAIAC | CoDA | CoDA-action |
|---|---|---|---|
| Runtime (min) | ~13 | ~10 | ~1 |

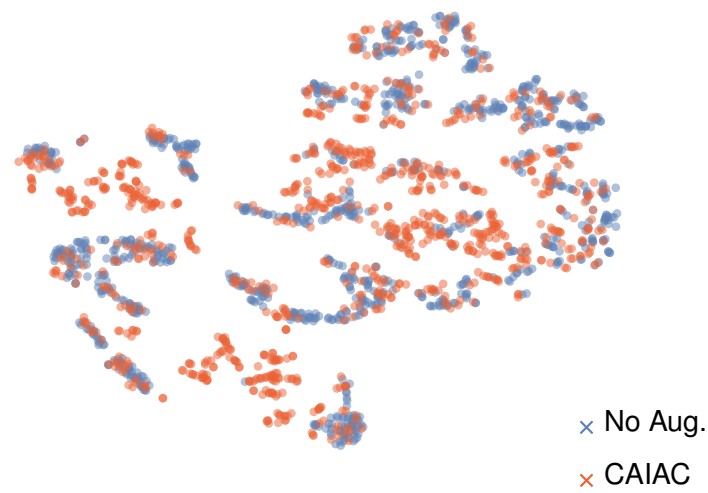

Figure 12: Original data and counteractuals augmentations visualized using t-SNE plot. For 1000 samples of the Franka-Kitchen dataset we add each one counterfactual augmentation with CAIAC.

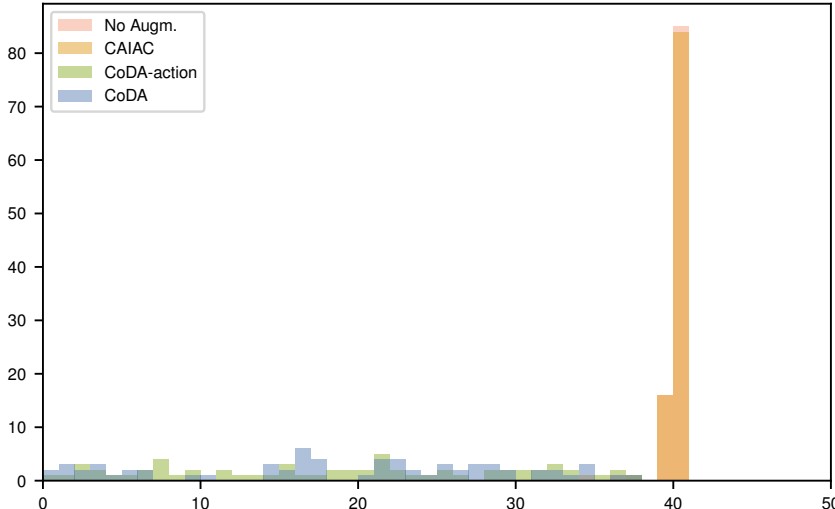

Figure 13: Log probabilities under the environment transition kernel for the counterfactuals skills created with the different methods. We see how CAIAC's augmented skills have similar log probabilites to samples from the real observed data.

