# OpenReview forum: "Causal Influence-Aware Counterfactual Data Augmentation"
_ICLR.cc/2024/Conference — Submitted to ICLR 2024_

### Official Review · Reviewer_LtoA · 2023-10-29

**Soundness:** 2 fair
**Presentation:** 3 good
**Contribution:** 2 fair
**Rating:** 3
**Confidence:** 4

**Summary:**

This paper emphasizes the importance of allowing learning agents to generalize to various situations, rather than being constrained by limited demonstrations. In real-world scenarios, the combinatorial complexity necessitates a significant amount of data to prevent neural network policies from relying on non-causal factors. Therefore, the authors propose CAIAC, a data augmentation method that generates synthetic samples from a fixed dataset without requiring new interactions with the environment. This method is inspired by the idea that an agent can only change its environment through actions. Hence, parts of the state space that are unaffected by actions are swapped from different trajectories in the dataset. The paper utilizes Causal Action Influence" (CAI) to identify action-independent entities and then swap states of these entities from other observations in the dataset. The introduction highlights the potential of teaching robots using demonstrations and datasets, which is a promising approach for developing competent robotic assistants.

**Strengths:**

1. The idea of using local independence to do counterfactual data augmentation is neat and interesting. As there could exist many spurious correlations during the offline data collection process, counterfactual data augmentation is important for breaking the spurious correlation.

2. The paper is well-written and easy to follow. The formulation of the problem and the proposed method is clear.

**Weaknesses:**

1. If I understand correctly, a strong underlying assumption of the proposed method is that the swapped states are irrelevant to the goal state. In the experiment part, the authors mention that “we initialize all non-target entities (with p = 0.5) to a random state”, which is why I think there exists such an assumption. This assumption is required since local independence does not imply the dependency between the current state and the goal state. This assumption is fine if the task is simple and the horizon is short. However, if the task is long-horizon and the later sub-tasks require some pre-conditioned to be satisfied, the local independence may not always be true.

2. Another assumption of the method is the fixed factorization of the state space, which may not be available in most real-world tasks. Determining which variable to be abstract from raw sensors may limit the usage of this method.

3. Missing related literature on causal reinforcement learning [1-11].

4. Using counterfactual data augmentation to improve RL algorithms has been investigated a lot in previous work. CoDA is an important work but cannot cover all existing baselines. The authors may need to add more baselines to show fair comparison, for example [1, 2, 4, 7, 9].


---
[1] Pitis, S., Creager, E., Mandlekar, A., & Garg, A. (2022). Mocoda: Model-based counterfactual data augmentation. NeurIPS 2022

[2] Lu, C., Huang, B., Wang, K., Hernández-Lobato, J. M., Zhang, K., & Schölkopf, B. (2020). Sample-efficient reinforcement learning via counterfactual-based data augmentation. arXiv preprint arXiv:2012.09092.

[3] Ding, W., Lin, H., Li, B., & Zhao, D. (2022). Generalizing goal-conditioned reinforcement learning with variational causal reasoning. NeurIPS 2022

[4] Ding, W., Shi, L., Chi, Y., & Zhao, D. (2023). Seeing is not believing: Robust reinforcement learning against spurious correlation. NeurIPS 2023

[5] Wang, Z., Xiao, X., Xu, Z., Zhu, Y., & Stone, P. (2022). Causal dynamics learning for task-independent state abstraction. ICML 2022

[6] Ke, N. R., Didolkar, A., Mittal, S., Goyal, A., Lajoie, G., Bauer, S., ... & Pal, C. (2021). Systematic evaluation of causal discovery in visual model-based reinforcement learning. arXiv preprint arXiv:2107.00848.

[7] Lyle, C., Zhang, A., Jiang, M., Pineau, J., & Gal, Y. (2021). Resolving causal confusion in reinforcement learning via robust exploration. In Self-Supervision for Reinforcement Learning Workshop-ICLR (Vol. 2021).

[8] Zhang, A., Lyle, C., Sodhani, S., Filos, A., Kwiatkowska, M., Pineau, J., ... & Precup, D. (2020, November). Invariant causal prediction for block MDPs. ICML 2020

[9] Zhang, A., McAllister, R., Calandra, R., Gal, Y., & Levine, S. (2020). Learning invariant representations for reinforcement learning without reconstruction. ICLR 2021

[10] Gasse, M., Grasset, D., Gaudron, G., & Oudeyer, P. Y. (2021). Causal reinforcement learning using observational and interventional data. arXiv preprint arXiv:2106.14421.

[11] Buesing, L., Weber, T., Zwols, Y., Racaniere, S., Guez, A., Lespiau, J. B., & Heess, N. (2018). Woulda, coulda, shoulda: Counterfactually-guided policy search. ICLR 2019

**Questions:**

1. The analysis of the failure cases in Table 1 is missing. The proposed method does not have an improvement in the last three tasks (i.e., Slide cabinet, Light switch, Hinge cabinet). I also observe such a performance drop in Figure 5. According to the design of the spurious correlation, I expect that the proposed method should generally work for all tasks. Could the authors explain the reasons for the failure?

2. In Section 5.2, the authors explore a goal-conditioned task. One question about the results is the statement “All methods perform similarly, given that there is enough coverage of the state space in the original dataset.”. It looks like the proposed method is the worst among all four methods. I don’t think the gap between CAIAC and No Aug. is caused by randomness. Usually, using data augmentation will not harm the performance. Could the authors provide some explanations? Is this related to the first point of the weakness part of my review?

3. Still in Section 5.2, the statement “the transformer model is able to discover the causal graph and creates realistic counterfactuals” is not supported by any evidence.

4. Could the authors provide a detailed comparison between CAIAC and CoDA? I think these two methods have very similar ideas but with different implementations. CoDA may suffer the problem of data scarcity for training a good transformer, but generally, what is the main advantage of using CAI to identify local independence?

5. “For Fetch-Push we set θ = 0.1, and θ = 0.3 for Franka-Kitchen.” How do you select the parameter θ?

---

> ### Author Response · Authors · 2023-11-22
> **Response to Reviewer LtoA (Part 1)**
>
> We thank the reviewer for their valuable feedback. We first address the points under weaknesses and then answer the reviewer’s questions. We address weaknesses with W# and questions with Q#.
>
>
> **W1: Assumption about independence swapped states and goal states:**
>
> We thank the reviewer for spotting a lack of detail in the manuscript. We would like to further elaborate on this point.
>
> We agree that in cases where demonstrations were provided to achieve desired goals, augmentations to a given state should be irrelevant to the goal that the trajectory achieves. In the Franka-Kitchen environment, we use the intersection of the uncontrollable sets along extended time windows, which excludes intermediate and final entities under influence that are potentially relevant for the demonstration. This is described in Equation 5) of the manuscript. We added a reference to section 3.3 as well.
>
> In addition, all three experiments (Franka-Kitchen, Fetch-Push and Fetch-Pick&Lift) rely on self-supervised goals: the training data does not provide an explicit goal space, nor desired goals per trajectory. For each experiment, goals for counterfactual states are relabeled by either sampling the final state of the skill (Franka-Kitchen), a random state from any trajectory (Fetch-Push), or a future state within the trajectory (Fetch-Pick&Lift). In any of these cases, as long as all trajectories are augmented before goal sampling, no assumption on independence between desired goals and states is necessary. We added clarifications on goal sampling in the updated manuscript (see Section 5.2,  A.1.1 and A.2.1).
>
> Moreover, to the best of our knowledge, the issue of dependence between states and goals is also less of a concern when performing offline reinforcement learning with goal relabeling (as done in Fetch experiments). In this case, a reward function can be used to evaluate success (as done in HER [F1]). Thus, a given trajectory does not need to successfully achieve its goal (in other words, the dependency between states and goals does not need to hold).
>
> Therefore, in Fetch experiments, the joint distribution over states and goals at training time only needs to assign a significant probability mass to state-goal pairs observed at test time. As goals are relabeled from augmented states, improved coverage over the joint state space at training time also results in improved coverage over the joint goal space. Finally, coverage over the space of state-goal pairs can be controlled through the relabeling strategy.
>
> [F1] Andrychowicz, Marcin, et al. "Hindsight experience replay." Advances in neural information processing systems 30 (2017).

---

> > ### Author Response · Authors · 2023-11-22
> > **Response to Reviewer LtoA (Part 2)**
> >
> > **W2: Assumption on factorization of the state space:**
> >
> > Our work does indeed rely on a fixed and known factorization, which might not be available in all scenarios. As already mentioned in the paper (see Section 2, paragraph 2), there is an important line of research for  methods that allow to automatically determine the number of factors and to learn latent representations of each entity [R1,R2, R3, R4, R5]. We added additional references to the updated version of the manuscript (see Section 2). Our method can be applied on top of such techniques. We would also like to  mention that such an assumption has been made in recently published papers such as [3, R6,R7,R8, R9, R10, R11].
> >
> > [R1] Francesco Locatello, Dirk Weissenborn,Thomas Unterthiner,Aravindh Mahendran,Georg Heigold,Jakob Uszkoreit, Alexey Dosovitskiy, and Thomas Kipf. Object-centric learning with slot attention. In NeurIPS, 2020.
> >
> > [R2]Klaus Greff,Raphael Lopez Kaufman,Rishabh Kabra,Nicholas Watters,Christopher P.Burgess ,Daniel Zoran, Loïc Matthey, Matthew M. Botvinick, and Alexander Lerchner. Multi-object representation learning with iterative variational inference. In ICML, 2019.
> >
> > [R3] Christopher P. Burgess, Loïc Matthey, Nicholas Watters, Rishabh Kabra, Irina Higgins, Matthew M. Botvinick, and Alexander Lerchner. Monet: Unsupervised scene decomposition and representation. ArXiv, 2019.
> >
> > [R4] Jiang, Jindong et al. “SCALOR: Generative World Models with Scalable Object Representations.” International Conference on Learning Representations (2019).
> >
> > [R5] Seitzer, Maximilian, et al. "Bridging the gap to real-world object-centric learning." arXiv preprint arXiv:2209.14860 (2022).
> >
> > [R6]Pitis, S., Creager, E., Mandlekar, A., & Garg, A. (2022). Mocoda: Model-based counterfactual data augmentation. NeurIPS 2022
> >
> > [R7] Pitis, Silviu, Elliot Creager, and Animesh Garg. "Counterfactual data augmentation using locally factored dynamics." Advances in Neural Information Processing Systems 33 (2020): 3976-3990.
> >
> > [R8] Sancaktar, Cansu, Sebastian Blaes, and Georg Martius. "Curious exploration via structured world models yields zero-shot object manipulation." Advances in Neural Information Processing Systems 35 (2022): 24170-24183.
> >
> > [R9] Maximilian Seitzer, Bernhard Schölkopf, and Georg Martius. Causal influence detection for improving efficiency in reinforcement learning. In Advances in Neural Information Processing Systems (NeurIPS 2021), December 2021.
> >
> > [R10] Zadaianchuk, Andrii, Georg Martius, and Fanny Yang. "Self-supervised reinforcement learning with independently controllable subgoals." Conference on Robot Learning. PMLR, 2022.
> >
> > [R11] Mambelli, Davide, et al. "Compositional multi-object reinforcement learning with linear relation networks." arXiv preprint arXiv:2201.13388 (2022).
> >
> > **W3: Related literature:**
> >
> > We thank the reviewer for pointing us to these works. Paper [11] is already cited and discussed in the submitted version.
> > We will add and discuss the remaining papers in a careful revision of our extended related work section.
> >
> > After checking these papers, we would like to state that our claims and novelty is not affected. There is no method in the offline setting that also creates counterfactuals without prior knowledge of the causal structure.
> > [1] is a follow-up of CoDA that assumes access to the true causal graph, hence does not address the problem we are tackling. [2 and 11] are using models to generate counterfactuals: [2] a GAN and [3] a forward prediction model. In the latter, only counterfactuals that are achievable within few steps are possible. [3] is looking into the online setting and active data collection for causal discovery. [4] The NeurIPS 2023 papers were not public before we submitted our paper. We will cite it as concurrent work. [5] considers learning a state abstraction using a causal forward model and not counterfactual data-augmentation. [7] is in the online setting and proposes an exploration algorithm which enables causal hypothesis-testing by interaction with the environment, which is not possible in the offline setting we are tackling. [8] is investigating how to obtain reduced causal graphs by discovering a block structure, [9] is a follow-up of [8] and works at the pixel-level. [6] is also considering learning from pixels and investigates representation learning algorithms. Finally [10] considers causal model learning from mixed offline and online experience.

---

> > > ### Author Response · Authors · 2023-11-22
> > > **Response to Reviewer LtoA (Part 3)**
> > >
> > > **W4: Baselines**
> > >
> > > We agree that more baseline would be better, however, of the ones you mention only [2] would be applicable in principle.
> > > [1] is a follow-up of CoDA which assumes access to the true causal structure. Thus, would not be an apple-to-apple comparison.
> > > [2] Using a generative model is an interesting alternative. We could not run this baseline during the rebuttal, because there is no code available. It is a bit unclear why a GAN would be able to generate counterfactuals that are actually out-of-distribution as it is the case here.
> > > [4] is clearly concurrent work. We could not have compared to this method
> > > [7] is a method developed for the online case. They propose an exploration algorithm that gathers data necessary to learn a causally correct policy. This requires additional interaction with the environment, which is not possible in the offline setting we are tackling.
> > > [9] works on the pixel level and has a different goal.
> > >
> > > **Q1: Failure case analysis**
> > >
> > > We thank the reviewer for suggesting to further elaborate on empirical results. We provide this discussion below, and integrated it in the updated version of the manuscript (Section 5.1.2 and Appendix A.2.1).
> > > As stated in the paper, “The results, shown in Table 1, are consistent with the challenging nature of this benchmark, as the evaluated tasks involve OOD settings in terms of states and actions”. More specifically, we found that only a subset of tasks is shown to be performed directly from the initial configuration in the training data.
> > > In particular, the training data does not contain state and action trajectories that show how to solve each of the 3 mentioned tasks from the initial robot joint configuration. Instead, these 3 tasks are only demonstrated after solving other tasks, and hence starting from other robot configurations. For this reason, even with perfect counterfactual data augmentation, these tasks remain challenging. Specifically, out of the 1200 demonstrations in the dataset, containing different task sequences, only 3 objects are shown to be manipulated from the initial robot configuration: 60% of the trajectories solve the microwave task first, 30% show the kettle task first and 10% show the bottom right burner first. This aligns with the relative performance achieved for those tasks. For the 3 remaining tasks, namely the slide cabinet, the light and the hinge cabinet, there is no demonstration shown directly from the initial configuration, and hence the low performance. Moreover, we remark that this issue adds up to the OOD nature of states and goals at test time, thus defining a fundamentally challenging benchmark.
> > >
> > > **I also observe such a performance drop in Figure 5:** In this scenario, we are analyzing the impact of the ratio of counterfactual augmented data on the performance. When setting the ratio to 1, all the data used for training are synthetic counterfactual data points. In this highly ablated scenario, any inaccuracies in counterfactual data creation will be fully exploited, without the possibility of being corrected by using a portion of real data. Also notice the high variance in this case. A similar effect was observed in the CoDA paper [R7]. We added these additional comments to the description of Figure 5.

---

> > > > ### Author Response · Authors · 2023-11-22
> > > > **Response to Reviewer LtoA (Part 4)**
> > > >
> > > > **Q5: How do you select the parameter θ?**
> > > >
> > > > We thank the reviewer for bringing this point up. First of all, we want to point out that the original text had a typo. In the original results, for Fetch-Push we used θ = 0.1 for CAIAC and θ = 0.3, for CoDA and CoDA action. For the Franka Kitchen, we used θ = 0.3. These values were chosen by a coarse grid search.
> > > >
> > > > Thanks to the reviewers comments we carried out a thorough and systematic analysis on the impact of the threshold parameter θ to the downstream task performance for the Fetch-Push task, and we updated the Figure 4 in the paper accordingly with the updated parameters.
> > > > We provide some details on the analysis below and integrated it in the updated version of the manuscript (Section A.5). Our updated pipeline consists of first training 10 different world models for each method. Afterward, we ran a grid search over the parameter θ. We ran 3 seeds for each of the 10 models and picked the value for θ that optimized the downstream average performance among the 10 models and the 3 seeds. The optimal parameters are:  θ = 0.05  for CAIAC and θ = 0.2 for CoDA and CoDA action for the Fetch-Push task. The optimization was done for the medium data regime and the optimal parameters were kept fixed for the high and low data regimes.
> > > >
> > > > This extensive analysis led to an improvement in CAIAC’s performance on the Fetch-Push task for all data regimes. The original results were with just a single model seed. Thus, it turns out that CoDA and CoDA action were slightly harmed. The reason is that the attention weights have a high variance between different training seeds, so a specific threshold parameter would need to be selected per seed.
> > > > To analyze the accuracy of causal influence detection by the different methods, we used a heuristic of influence using domain knowledge similarly than in the CoDA paper [7] (mainly, the agent is assumed to have no influence on the object if more than 7cm apart). We show the ROC curves for action influence for both CoDA-action and CAIAC in Figure 9, 10 and 11 in the Appendix A.5. The figures suggest that transformer weights cannot provide a reliable detection even with a threshold tuned to each seed.
> > > > The same procedure was followed for the Fetch-Pick&Lift task, a new task added in the updated version of the paper (see Section 5.2), as suggested by reviewer GhVi.
> > > >
> > > >
> > > > **Q2: Performance in Goal-conditioned Task, Section 5.2:**
> > > >
> > > > As mentioned in the point above, we updated the results for the Fetch-Push task in Figure 4 (left). Using the optimized thresholds for each of the methods we can see that in the high data regime CAIAC performs in par with the No-Augmented baseline and significantly better than CoDA and CoDA-action. With the updated results the gap in performance is even more noticeable for the low and medium data regimes. We updated the text accordingly.
> > > >
> > > >
> > > > **Q3: transformer model not being able to create realistic counterfactuals:**
> > > >
> > > > We thank the reviewer for pointing this out. We further validated this hypothesis by visualizing the attention weights of the trained transformer. The results are shown in Figure 8 of the Appendix and suggest that the transformer model’s ability for causal discovery is generally flawed in this setting. In addition, we make a quantitative analysis of the causal influence detection in Fetch-Push in Appendix A.5.

---

> > > > > ### Author Response · Authors · 2023-11-22
> > > > > **Response to Reviewer LtoA (Part 5)**
> > > > >
> > > > > **Q4: Detailed comparison between CAIAC and CoDA:**
> > > > >
> > > > > The algorithms differ on two aspects: (1) the extent of causal discovery and (2) the methodology used for causal discovery.
> > > > > CoDA performs (1) full causal discovery, i.e. it tries to detect influence between all the entities in the causal graph, and (2) uses the attention weights of a trained transformer to infer causation.
> > > > > CAIAC performs (1) partial causal discovery, i.e. it only tries to detect the agent’s _action-influence_ on the entities, and (2) uses state-conditioned mutual information (namely CAI)  to infer causation.
> > > > > The novelty of the method is therefore not limited to the mechanism used to infer causation but should also include the assumption of sparsity in the causal graph outlined in Section 3.1 of the paper, which allows us to reduce the hard problem of local causal discovery to the more approachable problem of local action influence detection, and hence motivates partial causal discovery.
> > > > >
> > > > > We further demonstrate these differences in the updated version of the paper. In Section A.5 (Figure 7 and 8) we visualize the computed influences for the Fetch-Push task for all the methods: CAIAC influence scores (namely CAI), CoDA influence scores (namely the transformer’s attention matrix) and the CoDA-action scores (namely the values of the attention matrix that correspond to the row “action” and columns {“object1, object2 or agent”}). This didactic example displays how using attention to detect influence can lead to misestimation (as in CoDA and CoDA-action), whereas CAIAC is better suited for detecting influence in this setting. Finally, we also provide an analysis of the accuracy of influence prediction for CAI and CoDA in Section A.5 (Figures 9,10 and 11) for the Fetch-Push task.

---

### Official Review · Reviewer_1iyX · 2023-10-31

**Soundness:** 3 good
**Presentation:** 2 fair
**Contribution:** 1 poor
**Rating:** 5
**Confidence:** 3

**Summary:**

This paper proposes CAIAC, a novel counterfactual data augmentation technique that generates additional data by swapping causally “action-unaffected” state dimensions of different transitions. CAIAC identifies action unaffected state dimensions using a Causal Action Influence (CAI) metric. Empirically, CAIAC outperforms other counterfactual data augmentation techniques (CoDA and CoDA-ACTION, sort of interpolation between CAIAC and CoDA) on offline Franka Kitchen tasks and a two-block FetchPush task.

**Strengths:**

1. The topic of counterfactual data augmentation is of general interest to the RL community, as many real-world tasks have local causal structures (as noted in the paper).
2. The paper is well-motivated, and I found the description of local causal models easy to follow.

**Weaknesses:**

1. There seems to be a fine distinction between CAIAC, CoDA, and CoDA-ACTION that that isn’t quite clear to me. My understanding is as follows:
* CoDA uses a learn local causal model (or a hard-coded heuristic) to identify locally independent state dimensions and then generates augmented transitions by swapping the locally independent state dimensions of observed transitions. The resulting augmented transitions.
* CAIAC is identical to CoDA but uses a CAI metric to identify locally independent state dimensions.
* I could not understand the difference between CoDA and CoDA-ACTION.


  I hope the authors can clear up my confusion on this matter. If my current understanding of CAIAC vs CoDA is correct, then algorithmic contribution of this work is limited. In any case, a figure that clearly illustrates the difference between CoDA, CODA-ACTION, and CAIAC would be immensely helpful towards understanding (1) the CAIAC algorithm and (2) the novelty of this work. It would also be helpful if the authors evaluated these algorithms on a simple, didactic toy task like the SpriteWorld task used CoDA.

2. Empirical results seem weak. In Table 1, CAIAC outperforms baselines with obvious significance in 3/6 tasks (Kettle, Microwave, Bottom-burner), and struggles in the remaining 3 tasks (Slide Cabinet, Light Switch, Hinge Cabinet). Since the algorithmic contribution seems limited, I would like to see CAIAC evaluated on additional tasks -- tasks that show some learning progress with CAIAC and in an online learning setting. Some possible tasks: FetchSlide, FetchPickAndPlace, FetchStack, or the analogous PandaGym tasks.

3. The paper states that CoDA and CODA-ACTION are (1) unable to recover the correct causal graph and (2) create dynamically infeasible data which harms performance, but there is no empirical evidence to support this claim. Given a dataset of augmented transitions {(s, a, r, s')}, the authors might consider validating claim (2) by initializing simulation to s, taking action a, and then checking if s' equals the simulators true next state. Then we could compute the probability that each algorithm generates feasible data and see if CoDA and CoDA-ACTION are more likely to generate such data than CAIAC. Claim (1) would then follow immediately -- if an algorithm generates a relatively large amount of infeasible data, then it surely has the wrong causal model.

1. It’s not immediately clear what CAIAC is doing from Figure 1. I suggest explicitly stating in caption or the figure itself what is being swapped and what augmented data is generated.

Other comments:

1. I found Figure 6 to be quite helpful in understanding the CAI scores. If possible, this figure would be a nice addition to the main paper.

2. When describing the local causal structure in the chosen benchmark tasks, it may be beneficial to concretely describe the structure. In particular, the agent's actions only affect an object if the agent is in contact with the object.

3. The authors may find the following references particularly relevant to this work:
* MoCoDA [1] is an extension of CoDA that enables a user to control the distribution of augmented data.
* GuDA [2] is a framework for generating expert-quality augmented data.

[1] MoCoDA: Model-based Counterfactual Data Augmentation. Pitis et. al, NeurIPS 2022.

[2] Corrado & Hanna. Guided Data Augmentation for Offline Reinforcement Learning and Imitation Learning. arXiv:2310.18247

**Questions:**

1. In the weaknesses section, I suggested additional online RL experiments. CAIAC, like CoDA, can be used in online learning too, correct?

---

> ### Author Response · Authors · 2023-11-22
> **Response to Reviewer 1iyX (Part 1)**
>
> We thank the reviewer for their valuable feedback. We first address the points under weaknesses and then answer the reviewer’s questions. We address weaknesses with W# and questions with Q#.
>
>
> **W1: Distinction between CAIAC, CoDA, and CoDA-ACTION.**
>
> The algorithms differ on two aspects: (1) the extent of causal discovery and (2) the methodology used for causal discovery.
>
> CoDA performs (1) full causal discovery, i.e., it tries to detect influence between all the entities in the causal graph, and (2) uses the attention weights of a trained transformer to infer causation. CoDA-action is a baseline we introduce. It performs (1) partial causal discovery,i.e. it only tries to detect the agent’s action-influence on the entities, and (2) uses the attention weights of a trained transformer to infer causation. Finally, CAIAC performs (1) partial causal discovery and (2) uses state-conditioned mutual information (namely CAI)  to infer causation. The novelty of the method is therefore not limited to the mechanism used to infer causation, but should also include the assumption of sparsity in the causal graph outlined in Section 3.1 of the paper, which motivates partial causal discovery.
>
> We further demonstrate these differences in the updated version of the paper. In Section A.5 (Figure 7 and 8) we visualize the computed influences for the Fetch-Push task for all the methods: CAIAC influence scores (namely CAI) , CoDA influence scores (namely the transformer’s attention matrix) and the CoDA-action scores (namely the values of the attention matrix that correspond to the row “action” and columns {“object1, object2 or agent”}). This didactic example displays  how using such an implicit measure to detect influence can lead to misestimation (as in CoDA and CoDA-action), whereas CAIAC is better suited for detecting influence in this setting.
>
> We updated the text in the updated version of the paper referring the reader to the appendix sections of interest to clarify the points above.
>
> **W2 + W3:  A figure that clearly illustrates the difference between CoDA, CODA-ACTION, and CAIAC [...]  + Simple, didactic toy task like the SpriteWorld task used CoDA**
>
> We hope that the explanations above helped clarify the distinctions between the different algorithms. We updated the text in the updated version of the paper referring the reader to the appendix where we include the computed influence scores mentioned in the point above. We hope that this scenario with only 3 entities (agent, object 1 and object 2) is already a good didactic example.
>
> **W4: Empirical results seem weak. In Table 1 [...] struggles in the remaining 3 tasks (Slide Cabinet, Light Switch, Hinge Cabinet).**
>
> We thank the reviewer for suggesting to further elaborate on empirical results. We provide this discussion below, and integrate it in the updated version of the manuscript (Section 5.1.2 and Appendix A.2.1).
>
> As stated in the paper, “The results, shown in Table 1, are consistent with the challenging nature of this benchmark, as the evaluated tasks involve OOD settings in terms of states and actions”. More specifically, we found that only a subset of tasks is shown to be performed directly from the initial configuration in the training data.
> In particular, the training data does not contain state and action trajectories that show how to solve each of the 3 mentioned tasks from the initial robot joint configuration. Instead, these 3 tasks are only demonstrated after solving other tasks, and hence starting from other robot configurations. For this reason, even with perfect counterfactual data augmentation, these tasks remain challenging.
>
> Specifically, out of the 1200 demonstrations in the dataset, containing different task sequences, only 3 objects are shown to be manipulated from the initial robot configuration: 60% of the trajectories solve the microwave task first, 30% show the kettle task first and 10% show the bottom right burner first. This aligns with the relative performance achieved for those tasks. For the 3 remaining tasks, namely the slide cabinet, the light and the hinge cabinet, there is no demonstration shown directly from the initial configuration, and hence the low performance.
>
>  Moreover, we remark that this issue adds up to the OOD nature of states and goals at test time, thus defining a fundamentally challenging benchmark.

---

> ### Author Response · Authors · 2023-11-22
> **Response to Reviewer 1iyX (Part 2)**
>
> **W5: [...]  would like to see CAIAC evaluated on additional tasks -- tasks that show some learning progress with CAIAC and in an online learning setting [...]**
>
> Following reviewer’s suggestions, we added an additional experiment: Fetch-Pick&Lift  with 4 blocks (see Figure 4 right), where there is a mismatch between the empirical joint state distribution in the training data and the joint state distribution at test time.
> On this task, CAIAC can create useful counterfactual samples and allows  generalization beyond the joint state distribution in the training data, thus outperforming other methods by 60%. The rest of the baselines cannot create the needed counterfactuals and have significantly lower performance.
>
> CAIAC is mainly aimed at settings in which the distribution and quantity of training data cannot be controlled, and distribution mismatch is thus a fundamental issue. While applying CAIAC to the online setting remains possible, the continual collection of on-policy data can already alleviate issues such as spurious correlation in the replay buffer or insufficient coverage of the joint state space. While data augmentation can also be helpful in online learning, we believe that centering our experiments on the offline setting can better isolate the issues the method is tackling, and constitutes a more meaningful benchmark.
>
> **W6: Empirical evidence on quality of counterfactuals**
>
> We appreciate the reviewer’s nice suggestion since we believe it provides additional empirical support to this work.
> First of all, we want to refer the reader to the Section A.4  of the updated version of the paper, where we show the attention weights of the trained transformer (see Figures 7 and 8) to provide evidence of incorrect causal estimation by CoDA and CoDA action. The weights are even unable to recover influence along the diagonal, i.e., that an entity state at time $t+1$ depends on the entity state at time $t$. These visual results give evidence for claim 1).
>
> **Comment 1: I found Figure 6 to be quite helpful in understanding the CAI scores.**
>
> We appreciate the comment. We added a reference to the figure in the main paper (Section 5.1.2) as well to an additional analogous figure for the Fetch-Push task (see Figure 7 in the Appendix) which also has an added reference in the main paper (see Section 5.2).
>
> **Comment 2: When describing the local causal structure in the chosen benchmark tasks, it may be beneficial to concretely describe the structure. In particular, the agent's actions only affect an object if the agent is in contact with the object.**
>
> We would like to clarify that through our work we adopt a unified definition of influence: the agent has influence over an entity in a specific state configuration if it can modify the state of the entity at next time step through its actions (see equation 3  for a formal definition). Concretely, in the case of robotic manipulation environments physical contact is not necessary as long as the agent can change the object pose, even if indirectly,  in a single simulation step. We clarified this in the updated version of the manuscript (see Appendix A4).
>
> **Comment 3: Related literature**
>
> We thank the reviewer for pointing us out to these papers. We will add and discuss the 2 mentioned papers in an extended related work section.
> After checking these papers, we would like to mention that [2] was not published yet by the time we submitted our manuscript and will be cited as concurrent work. Furthermore [1] is an interesting follow-up of CoDA but assumes access to the true causal graph, and hence does not address the problem we are tackling.
>
> **Q1: Additional online RL experiment**
>
> Please see answer to weakness W5 above.

---

> ### Author Response · Authors · 2023-11-23
> **Response to Reviewer 1iyX (Part 3)**
>
> **W6: Empirical evidence on quality of counterfactuals (Part 2)**
>
> We provide an analysis on the created counterfactuals using CAIAC.
> We visually show how the created augmented samples have an increased support of the joint state space distribution in the kitchen dataset, compared to the No-Augmented baseline.
> In Figure 12 of the updated manuscript,  we show 1000 randomly selected samples from the Franka-Kitchen dataset together with one counterfactual for each. The visualization employs t-SNE for a 2D representation of the high-dimensional state space.
> We observe how the augmented samples using CAIAC cover a much larger space than the observed data, suggesting that we are able to provide states with a richer amount of configurations.
> However, this does not tell us whether these counterfactuals are actually valid.
> To quantify whether an augmented datapoint is valid, we do the following:
> We consider time-windows  $(s_t, \dots, s_{t+\tau})$ sampled from the Franka-Kitchen dataset.
> We create counterfactuals for this window with \method and CoDA, $(\tilde s_t, \dots,\tilde s_{t+\tau})$.
> We reset the simulator to $\tilde s_t$ and simulate the future with the action sequence in the original trajectory for $\tau$ steps.
> If the counterfactual is valid, then resulting state of the simulation should coincide with the counterfactual $\tilde s_{t+\tau}$.
> Unfortunately, the simulator is not perfectly deterministic (or we are unable to set the full simulator state), so measuring exact matching states is not possible.
> Thus, we run the simulator $K=10$ times with different env-seeds and obtain $\mathcal S' = [ \bar s^k_{t+\tau} ]_{k=1}^K$.
> We fit a multivariate Gaussian to the set $\mathcal S'$.
>
> We now compute the probability of the counterfactual $\tilde s_{t+\tau}$ under the Gaussian distribution. High probability means valid counterfacturals, low means invalid.
> In Figure 13 of the updated manuscript we present the histogram of log probabilities. The counterfactuals by CAIAC are mostly valid, which is not the case for the CoDA baselines.

---

### Official Review · Reviewer_GhVi · 2023-11-08

**Soundness:** 2 fair
**Presentation:** 3 good
**Contribution:** 2 fair
**Rating:** 5
**Confidence:** 3

**Summary:**

This paper proposes a method called Causal Influence Aware Counterfactual Data Augmentation (CAIAC) that addresses the challenge of generalizing robot behaviours to new situations using pre-recorded data and human-collected demonstrations. By swapping causally action-unaffected parts of the state-space from different observed trajectories in the dataset, CAIAC creates feasible synthetic samples without the need for new environment interactions. The experimental results demonstrate the generalization capabilities and sample efficiency of the proposed method.

**Strengths:**

- The paper proposes a data augmentation method called Causal Influence Aware Counterfactual Data Augmentation (CAIAC) that can create feasible synthetic samples from a fixed dataset without the need for new environmental interactions.

- The paper is well-written and easy to follow.

- The experiments on offline self-supervised skill learning and offline reinforcement learning showcase the effectiveness of the proposed method as some extent.

- The proposed approach is independent and can be used with any learning algorithm.

**Weaknesses:**

- The novelty is limited, drawing heavily on the groundwork laid by Seitzer et al., 2021, for local causal graph estimation and CAI's influence measurement. The conceptual leap from the work of CoDA (Pitis et al., 2020), which also involves counterfactual generation through connected component swapping, to the present technique of swapping uncontrollable subgraphs, seems incremental rather than revolutionary.

- While the paper successfully argues the challenges and pitfalls of complete causal structure estimation, it only partially addresses the performance of CAIAC in high-dimensional, low data regime environments, leaving a gap in the analysis. A more exhaustive exploration of the method's computational demands and scalability would greatly enhance the reader's understanding.

- The experimental comparisons seem to lack a critical control condition — an alternative method that also augments counterfactual data through local causal structure estimation with CAI but swap the connected components to form new transitions given two transitions that share local causal structures. Including such a benchmark would provide a clearer picture of CAIAC's relative efficacy.

**Questions:**

- Could the authors provide more insight into CAIAC's performance in environments with abundant data? The discrepancy in performance between low and high data regimes in high-dimensional settings warrants further clarification.

- Moreover, could the authors elaborate on the computational complexity and scalability of the CAIAC method, especially in comparison to existing methods?

---

> ### Author Response · Authors · 2023-11-22
> **Response to Reviewer GhVi (Part 1)**
>
> We thank the reviewer for their valuable feedback. We first address the points under weaknesses and then answer the reviewer’s questions. We address weaknesses with W# and questions with Q#.
>
> **W1: Novelty:**
> To the best of our knowledge, this is the first proposed method for data augmentation that uses an explicit measure of influence between entities in the environment to create counterfactuals. While CoDA also suggests creating counterfactuals by swapping local independent subgraphs from different transitions, it either relies on hardcoded dependencies, or uses a heuristic upon the attention weights of a transformer world model to infer the local factorization. As we show, such a heuristic is generally not accurate, and leads to the creation of unfeasible augmented samples that hurt performance. In the updated version of the paper, we provide evidence by showing a comparison between the different measures of influence on the Franka Kitchen and Fetch-Push tasks (Section A.4, Figures 6, 7 and 8), and an analysis of the accuracy of influence prediction for CAI and CoDA in Section A.5 (Figures 9,10 and 11) for the Fetch-Push task.
>
> **W2: Performance difference/ gap in analysis**:
> To rephrase, the reviewer is asking to motivate the difference in performance between CoDA and CAIAC. To analyze this, we designed the baseline CoDA-action, which aims at discovering the same part of the causal structure as CAIAC, but using the aforementioned heuristic of transformer attention weights. The fact that CoDA-action performs comparable to CoDA suggests that the assumption of sparsity in the causal graph is not sufficient, but the more accurate measure of causal influence is crucial.
>
> **W3: Computational demands:**
> CAIAC relies on computing the CAI measure for data augmentation. In turn, CAI can be evaluated for all entities at once, through $k$ forward passes for the $k$ counterfactual actions, which are performed in a batch-wise fashion.
> $k$ is a constant factor, and does not scale with the number of entities.
> Methods relying on a transformer world model, like CoDA and CoDA-action only need one forward pass (which internally has quadratic cost in the number of entities due to cross attention). However, CoDA also needs to compute the connected components from the adjacency matrix, which has a quadratic cost.
> For relatively few entities, as is common in the robotic manipulation environments, the computational overhead is relatively small.  For the high data regime Fetch-Push environment we timed how long it takes for each algorithm to compute influence on all 2M datapoints:
>
> | Method      | Runtime     |
> |-------------|-------------|
> | CoDA-action | 47 seconds  |
> | CoDA        | ~10 minutes |
> | CAIAC       | ~13 minutes |
>
>
> The algorithms were benchmarked on a 12-core Intel i7 CPU.
> We would like to mention that counterfactuals could be generated in parallel to the learning algorithm and hence not significantly impact runtime of the algorithm. Furthermore, in our offline setting, counterfactuals can be fully precomputed.
> This information was added in the updated version of the manuscript (see Appendix A6).
>
>
> **W4: Control condition: swapping connected components:**
> To rephrase, the reviewer is suggesting to create counterfactuals by swapping the controllable subgraphs instead of the uncontrollable ones. Both approaches are correct and would lead to feasible counterfactuals. However, the coverage of the state space is greater using the current approach. In the case that given a specific environment configuration only one task is demonstrated (such as in the kitchen), swapping the controllable subgraphs would lead to counterfactuals that are likely already part of the dataset.

---

> ### Author Response · Authors · 2023-11-22
> **Response to Reviewer GhVi (Part 2)**
>
> **Q1:High data regime performance:**
>
> This is a good point. We have updated the results on the Fetch-Push experiment (Figure 4, left) and added a new experiment on a Fetch-Pick&Lift environment (Figure 4, right). The updated results stem from a more systematic hyperparameter tuning for **each** method (see reviewer LtoA (Part 5, Q5) and our answer there). While the results follow a similar trend to the original one, it notably demonstrates significantly higher performance for CAIAC compared to other methods.
> In order to further analyze the obtained results, we would like to consider three conditions separately. We also added these clarifications in the updated version of the manuscript (see Section 5.2).
>
> 1. When data is scarce (see Fetch-Push, low-data regimes), as long as the marginal state distributions at training and test time match for all the entities, CAIAC is able to cover the full support of the state joint distribution by creating unseen configurations of the entities.
> 2. When data is abundant AND it adequately covers the full joint state space (as in Fetch-Push, high-data regime), no data augmentation is needed by definition and hence CAIAC and the No-Augmented baseline perform similarly. For CoDA and CoDA action we see a light drop in performance possibly due to the creation of unfeasible counterfactuals which harm performance.
> 3. When data is abundant AND there is yet a mismatch between the joint state distribution at training and test time, CAIAC remains very effective. We showcase this in a new experiment on Fetch-Pick&Lift (see Figure 4 right). We use abundant data, namely 2M samples/40k episodes. However, the state joint distribution in the training data does not match the state joint distribution at test time. CAIAC reaches almost 100% whereas all baselines perform purely. CoDA cannot create helpful counterfactuals, even with an optimized detection threshold.
>
> **Q2: Computational demands:**
>
> Please see answer to W3 point above.

---

### Author Response · Authors · 2023-11-22
**Common response**

We would like to thank the reviewers for their time and effort in reviewing our paper.
By addressing their comments, we believe we have significantly enhanced the quality of our manuscript. We provide a summary of the main changes made in the updated version. For details, we refer to the individual responses to each reviewer.

**1. Difference between CAIAC and CoDA baseline: (Reviewer GhVi, Reviewer 1iyX, Reviewer LtoA)**. We updated the manuscript to clarify the following points. The algorithms differ in two aspects: (1) the extent of causal discovery and (2) the methodology used for causal discovery. CoDA baseline performs (1) full causal discovery (i.e. it tries to detect influence between _all_ the entities in the causal graph) and (2) uses an implicit measure to detect causation (namely the attention weights of a transformer). In contrast, CAIAC performs (1) partial causal discovery (i.e. it tries to detect agent’s _action-influence_ on the entities) leveraging a causal graph sparsity assumption which allows to (2) use an explicit measure to infer action causation (namely state-conditioned mutual information).

**2. New experiment: (Reviewer 1iyX).** We include a new experiment on a Fetch-Pick&Lift environment with 4 blocks. The experiment is in a high data regime, but there is a mismatch between the empirical joint state distribution in the training data and the joint state distribution at test time. On this task, CAIAC can create useful counterfactual samples that allow generalization beyond the joint state distribution in the training data and achieve almost 100% success rate. All baselines show a significant lower performance.

**3. Report on influence scores: (Reviewer 1iyX, Reviewer LtoA).** We provide a thorough analysis of the computed influence measures for different experiments and for all methods. Given the results, we observe  why the current baselines are unable to discover the correct causal graph and as a consequence, fail to generate realistic augmented samples.

**4. Systematic optimization of the threshold influence parameter: (Reviewer LtoA)**. Over the course of the discussion period, we carried out a thorough optimization of the threshold parameters for all methods. This led to improvements in results for the Fetch-Push task which were updated in the new version of the manuscript.

For an easier visualization of the changes made in the updated manuscript, we temporarily colored all the added new text in red.
If there are remaining unclear points, we would be happy to clarify them.

---

### Meta-Review · Area_Chair_aida · 2023-12-06

**Metareview:**

This paper introduces CAIAC, a data augmentation method designed to enhance the learning of complex robot behaviors using a fixed dataset. CAIAC creates synthetic samples without new environment interactions by swapping causally unaffected parts of the state-space from different observed trajectories. This approach addresses the combinatorial challenges of real-world scenarios, promoting increased generalization capabilities and sample efficiency in high-dimensional benchmark environments.

While the proposed idea has some novelty, the technical contribution compared to existing works are not substantial and the empirical results are not strong. In addition, the presentation needs improvement and the comparison to existing methods are not sufficiently discussed. I would suggest further improving the manuscript according to the review comments and submit in future venues.

**Justification For Why Not Higher Score:**

technical contribution not sufficient

**Justification For Why Not Lower Score:**

n/a

---

### Decision · Program_Chairs · 2024-01-16

Reject